# 2-deoxyglucose transiently inhibits yeast AMPK signaling and triggers glucose transporter endocytosis, potentiating the drug toxicity

Clotilde Laussel[ID][ʘ], Véronique Albanèse[ID][ʘ], Francisco Javier García-Rodríguez[ID][‡], Alberto Ballin[ID][‡], Quentin Defenouillère[ID], Sébastien Léon[ID]*

Université Paris Cité, CNRS, Institut Jacques Monod, Paris, France

ʘ These authors contributed equally to this work.
‡ FG and AB also contributed equally to this work.
* sebastien.leon@ijm.fr

**Data Availability Statement:** All relevant data are within the manuscript and its Supporting Information files.

## Abstract

2-deoxyglucose is a glucose analog that impacts many aspects of cellular physiology. After its uptake and its phosphorylation into 2-deoxyglucose-6-phosphate (2DG6P), it interferes with several metabolic pathways including glycolysis and protein N-glycosylation. Despite this systemic effect, resistance can arise through strategies that are only partially understood. In yeast, 2DG resistance is often associated with mutations causing increased activity of the yeast 5'-AMP activated protein kinase (AMPK), Snf1. Here we focus on the contribution of a Snf1 substrate in 2DG resistance, namely the alpha-arrestin Rod1 involved in nutrient transporter endocytosis. We report that 2DG triggers the endocytosis of many plasma membrane proteins, mostly in a Rod1-dependent manner. Rod1 participates in 2DG-induced endocytosis because 2DG, following its phosphorylation by hexokinase Hxk2, triggers changes in Rod1 post-translational modifications and promotes its function in endocytosis. Mechanistically, this is explained by a transient, 2DG-induced inactivation of Snf1/AMPK by protein phosphatase 1 (PP1). We show that 2DG-induced endocytosis is detrimental to cells, and the lack of Rod1 counteracts this process by stabilizing glucose transporters at the plasma membrane. This facilitates glucose uptake, which may help override the metabolic blockade caused by 2DG, and 2DG export—thus terminating the process of 2DG detoxification. Altogether, these results shed a new light on the regulation of AMPK signaling in yeast and highlight a remarkable strategy to bypass 2DG toxicity involving glucose transporter regulation.

## Author summary

In this article, we study the resistance to a drug named 2-deoxyglucose (2DG). 2DG is an efficient inhibitor of several metabolic pathways, particularly glycolysis which is of prime importance for tumor cell proliferation. Cancer cells but also cells from other organisms

**Funding:** This work was supported by grants from IDEx Université de Paris (https://u-paris.fr; ANR-18-IDEX-0001 to SL), the Fondation ARC pour la recherche sur le cancer (https://www.fondation-arc.org, ARCPJA32020060002096 to SL), and individual fellowships from the Ligue contre le cancer (https://www.ligue-cancer.net; TAZK20115 to CL), the Fondation pour la Recherche Médicale (https://www.frm.org; SPF20150934065 to QD) and the Ministère de l'Enseignement Supérieur et de la Recherche (https://www.enseignementsup-recherche.gouv.fr; to AB). The funders had no role in study design, data collection and analysis, decision to publish, or preparation of the manuscript.

**Competing interests:** The authors have declared that no competing interests exist.

can develop various resistance strategies that only being unraveled. Here, using baker's yeast as a model organism, we document the resistance mechanism of a mutant lacking the arrestin gene *ROD1*, known to regulate nutrient transporter endocytosis. First, we show that 2DG triggers the endocytosis of many nutrient transporters. Mechanistically, 2DG leads to a transient inhibition of the AMPK signaling pathway, which in turn promotes the activation of Rod1, a pre-requisite for 2DG-induced endocytosis. Consequently, cells lacking Rod1 maintain various proteins at the plasma membrane including hexose transporters, the latter being essential for 2DG resistance. Hexose transporter stabilization at the cell surface may sustain glucose import and help cells cope with the 2DG-induced metabolic blockade, while also allowing 2DG detoxication.

## Introduction

2DG is a glucose analog which lacks a hydroxyl group in position 2, and mainly acts as a competitive inhibitor of glucose metabolism [1]. As such, it competes with glucose for entry into cells and for its subsequent phosphorylation by hexokinase, but is almost not further metabolized. This leads to the intracellular accumulation of 2DG-6-phosphate, a toxic metabolite that inhibits glycolysis and thus leads to a rapid depletion of ATP stores, and the activation of nutrient starvation-activated pathways such as 5'-AMP activated protein kinase, AMPK [2]. Moreover, since mannose is the C2 epimer of glucose, 2-deoxyglucose is also 2-deoxymannose and thus also interferes with mannose metabolism, notably inhibiting protein N-glycosylation (reviewed in [3]). This causes a series of cellular effects, and triggers the onset of the Unfolded Protein Response pathway to face the stress encountered by a defective glycosylation of proteins at the endoplasmic reticulum, which is reportedly the main toxic effect of 2DG in some conditions [4].

The realization that many cancer cells display an aberrantly high glucose uptake and metabolism has led to the idea that glycolysis inhibitors could be used to preferentially target and kill tumor cells [5]. This also allowed the use of radiolabeled glucose derivatives, such as 18-fluoro-2-deoxyglucose, to preferentially label tumors for cancer imaging (PET-scans) [6]. So far, trials using 2DG as the sole chemotherapeutic agent have failed because of the relative toxicity of this molecule, however strategies using 2DG or other 2-deoxy-substituted versions of glucose in combination with other drugs are under study and are promising [6].

2DG has also been extensively used to understand glucose signaling in micro-organisms such as the baker's yeast *Saccharomyces cerevisiae*, where it has been instrumental to delineate the molecular mechanisms by which glucose regulates gene expression (reviewed in [3]). The metabolism of *S. cerevisiae* is biased towards the preferential use of glucose as a carbon source. At the molecular level, glucose represses the expression of many genes involved in respiratory metabolism or in the use of other carbon sources, thereby favoring the preferential use of glucose by fermentation [7]. Among the actors in charge of adapting the transcriptional program with respect to glucose availability is the yeast orthologue of AMPK, Snf1 [7,8]. Snf1 is the catalytic subunit of a heterotrimeric complex which is active in the absence of glucose in the medium. A canonical example of a Snf1-regulated transcription is that of the invertase *SUC2*, required for sucrose hydrolysis and metabolism. Transferring yeast cells to sucrose medium activates Snf1, causing the phosphorylation of the transcriptional repressor Mig1, its translocation out of the nucleus and the derepression of *SUC2* [9,10].

Snf1, like other kinases of the AMPK family, is activated by phosphorylation of a residue in its activation loop [11]. This involves one of 3 upstream kinases whose activity is, surprisingly,

not regulated by glucose. Rather, evidence suggests that Snf1 activity is regulated by its dephosphorylation by the Protein Phosphatase 1 (PP1) complex made of the catalytic subunit, Glc7, and the glucose-specific regulatory subunit Reg1 [12]. How PP1 activity is linked to glucose availability is not yet clear, but deletion of *REG1* leads to constitutive Snf1 activity and to the expression of glucose-repressed genes even in the presence of glucose [13].

Another phenotype displayed by the *reg1Δ* strain is its ability to grow robustly in presence of 2DG [3,14,15]. This is attributed to the constitutive activation of Snf1 in this mutant, as mutations in *SNF1* cause hypersensitivity to 2DG and yeast lacking both Reg1 and Snf1 are also hypersensitive [16].

Despite the multifaceted cellular effects of 2DG, cells can indeed overcome 2DG toxicity by mechanisms that are currently being unraveled (reviewed in [3,15]). A prolonged exposure of HeLa cells in 2DG-containing medium gave rise to resistant clones [17]. These cells displayed a high phosphatase activity towards 2DG-6-phosphate (2DG-6-P), suggesting the presence of an enzyme that could detoxify this metabolite back into 2DG and thus prevent the metabolic blockade. Studies in yeast also identified phosphatases, named Dog1 and Dog2, with a similar activity *in vitro* and whose overexpression allows growth in 2DG-containing medium [18,19]. Recently, we found that the most abundant isoform, Dog2, is itself a glucose-repressed gene and thus is regulated by Snf1/AMPK function [20]. This partially explains why strains with high Snf1/AMPK activity are resistant to 2DG, such as *reg1Δ* [14,16], other *reg1* mutant alleles [20,21], a viable *glc7* mutation (Q48K) [21] or gain-of-function mutations in components of the AMPK complex [16,21,22]. However, additional resistance mechanisms independent of *DOG2* expression are at stake in these strains [20,21]. This is suggested by the observation that *DOG2* deletion does not completely re-sensitize *reg1Δ* mutants to 2DG, and a point mutation in *REG1* (*reg1-P231L*) confers 2DG resistance but does not impact on the glucose-mediated repression of genes [21]. Thus, Snf1 likely controls additional factors participating in 2DG resistance.

One such candidate is the arrestin-related protein, Rod1. Arrestin-related proteins (ARTs) are important regulators of plasma membrane protein endocytosis in response to extracellular cues [23,24]. ARTs recruit the ubiquitin ligase Rsp5 to nutrient transporters at the plasma membrane, promoting their subsequent ubiquitylation and endocytosis [25]. ART activity is regulated by nutrient signaling pathways, allowing to remodel the landscape of transporters at the plasma membrane to meet the physiological needs of cells facing a nutrient challenge [26,27]. Particularly, Rod1 regulates endocytosis in response to glucose availability, and its activity is oppositely regulated by Snf1 and PP1 [28–30], being a direct target of Snf1 [31]. A genome-wide screen revealed that *rod1Δ* is indeed resistant to 2DG [14], thus the Snf1-mediated inhibition of Rod1 might contribute to 2DG resistance.

Point mutations aimed at abolishing Rod1 interaction with the ubiquitin ligase Rsp5, thereby annihilating their function as adaptor proteins in endocytosis, increased resistance to 2DG suggesting that it mediates 2DG toxicity through its function in endocytosis [32]. 2DG resistance of the *rod1Δ* strain is further increased by the additional deletion of the *ROD1* paralogue *ROG3* [32]. Rod1 (and Rog3, to a lesser extent) controls the endocytosis of the glucose transporters Hxt1 and Hxt3 triggered in response to 2DG exposure, and overexpression of the same transporters in a *snf1Δ* mutant restored partial resistance to 2DG [32]. Deletion of *ROD1* and *ROG3* stabilized hexose transporters at the plasma membrane in *snf1Δ* and restored its 2DG-sensitivity to WT levels [32]. However, whether the effects of 2DG on endocytosis extend beyond glucose transporters is unknown, and there is currently no molecular understanding of how transporter stability/localization contributes to 2DG resistance.

In this study, we examined the effects of 2DG on endocytosis and found that 2DG triggers the endocytosis of many plasma membrane proteins, most of which depend on Rod1.

Mechanistically, we show that opposite to the situation in mammalian cells, 2DG treatment leads to the dephosphorylation of Snf1/AMPK and several targets including Rod1 in a-PP1 dependent manner. This favors Rod1 function and thus provides a rationale as to why this arrestin is central to 2DG-induced endocytosis. Finally, we demonstrate that 2DG resistance of the *rod1Δ* strain relies on the maintenance of active glucose transporters at the membrane, where they can increase glucose uptake and contribute to detoxifying 2DG out of the cells, thus terminating the detoxification process.

## Results

### 2DG triggers the endocytosis of many plasma membrane proteins

We previously described the proteomic changes occurring in response to 2DG and focused on proteins upregulated in this condition [20]. In addition, a few transmembrane proteins of the plasma membrane displayed a significantly decreased abundance after 2h30 2DG treatment, including the amino acid permease Tat1 and the hexose transporter Hxt2 [20]. This could either be caused by a decreased synthesis or an active degradation by endocytosis. We verified this by looking at the regulation of the endogenously GFP-tagged proteins. Treatment of glucose-grown cells with 0.2% 2DG led to the targeting of plasma membrane-localized Tat1 and Hxt2 (**Fig 1A**) to the vacuole, which was accompanied by their degradation (**Fig 1B**). Of note, we observed that the vacuole was strongly fragmented in response to 2DG (**S1 Fig**), confirming recent findings [33]. This may be due to the fact that 2DG causes ER stress in yeast [20], which itself induces vacuolar fragmentation [34].

Tat2 and Hxt2 endocytosis in response to 2DG is reminiscent of the behavior of the low affinity hexose transporters Hxt1 and Hxt3 [32]. To document the extent of plasma membrane remodeling caused by 2DG exposure, we studied the regulation of a representative subset of plasma membrane proteins tagged with GFP at their endogenous loci before and after 2DG treatment. This included transporters of various substrates or structural classes, as well as receptor/sensor proteins whose expression could be detected in glucose-grown cells. This showed that many of these membrane proteins are endocytosed in response to 2DG (**Figs 1D** and **S2**).

In yeast, nutrient transporter endocytosis is driven by their ubiquitylation at the plasma membrane, which relies on the action of the ubiquitin ligase Rsp5 and adaptor proteins of the arrestin-like family (ARTs) [24, 27]. Previous work highlighted the importance of the arrestin-related protein Rod1 in the 2DG-induced endocytosis of Hxt1 and Hxt3 [32]. Indeed, we found that most of the cargoes studied here depended on Rod1 for their 2DG-induced endocytosis (**Figs 1D** and **S2**). Although the Rod1 paralogue Rog3 displays some level of functional redundancy with Rod1 [32], Rog3 was not responsible for the endocytosis of the Rod1-independent cargo Lyp1 (**S3 Fig**), although this may be the case for other cargoes. Altogether, these results indicate a prominent role for the ART protein Rod1 in 2DG-induced endocytosis.

### 2DG triggers endocytosis by inducing Rod1 dephosphorylation and ubiquitylation

We then focused on understanding why Rod1 function is so central for 2-deoxyglucose endocytosis. In the case of glucose-induced endocytosis, Rod1 activity is regulated through changes in its post-translational modifications [28,29]. In a glucose-deprived medium, AMPK/Snf1 is active and phosphorylates Rod1, inhibiting its function [28,29,31,35]. Conversely, glucose addition in the medium leads to Rod1 dephosphorylation in a PP1-dependent manner, followed by its ubiquitylation by Rsp5 [28,29]. These modifications promote its activity as an

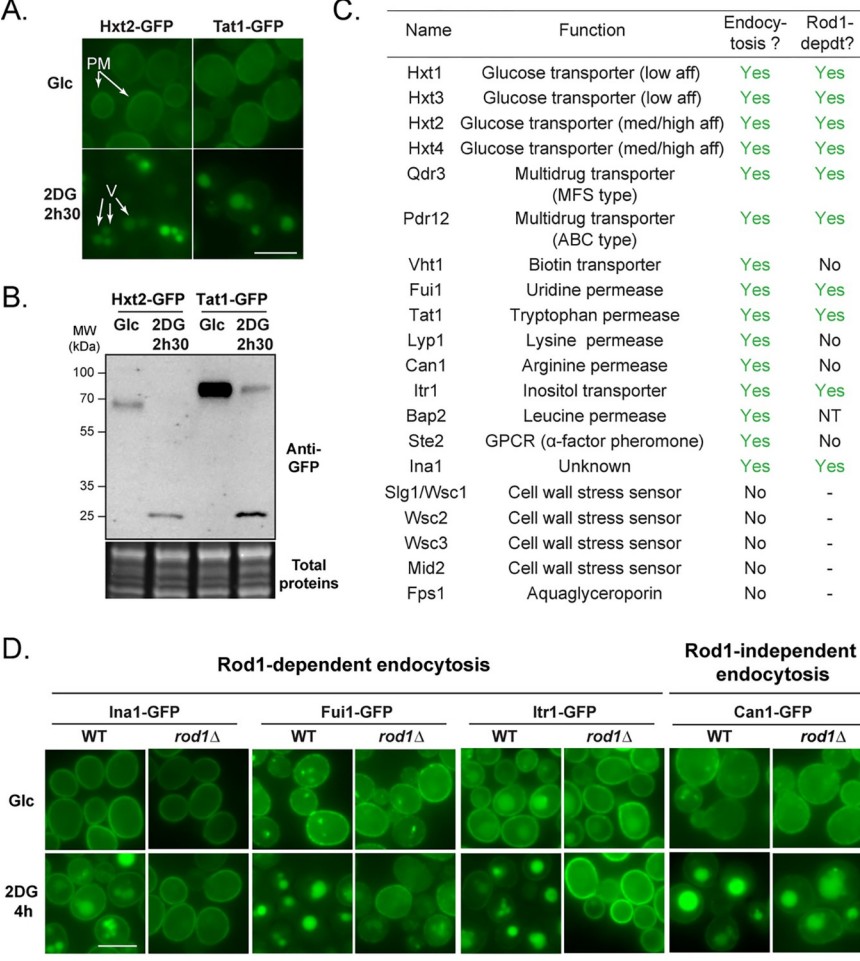

**Fig 1. 2DG treatment induces the endocytosis of many plasma membrane proteins, that is mostly Rod1-dependent.** (**A**) Cells expressing Hxt2-GFP and Tat1-GFP cells were grown overnight in a glucose-containing medium (exponential phase) and treated with 2DG for 2h30 and observed by fluorescence microscopy. PM: plasma membrane; V: vacuole. Scale bar: 5 μm. (**B**) Western blot on total protein extracts from Hxt2-GFP- and Tat1-GFP- expressing cells before and after 2DG addition for 2h30, using anti-GFP antibodies. (**C**) Plasma membrane proteins whose localization was examined by fluorescence microscopy before and after 2h30 2DG. When endocytosed, the same proteins were also observed in a *rod1Δ* context to check for Rod1 dependence (NT, not tested). (**D**) Fluorescence microscopy of WT and *rod1Δ* cells expressing the indicated transporters tagged with GFP before and after 2DG treatment for 4h. Scale bar: 5 μm.

Rsp5 adaptor, leading to transporter ubiquitylation, endocytosis and vacuolar sorting [28,29,35]. Similarly, 2DG addition to glucose-grown cells triggers Rod1 ubiquitylation in an Rsp5- and PP1-dependent manner, but paradoxically 2DG was also reported to increase Snf1 activity [30,32]. Thus, how 2DG regulates the relative activities of Snf1/PP1 and how this impacts on Rod1 activity are not fully understood.

Previously, the effect of 2DG on Rod1 post-translational modifications was evaluated using an 3HA-tagged construct [32]. Although Rod1-3HA could support the glucose-induced endocytosis of Jen1 [28], we realized that it was unable to restore the 2DG-induced endocytosis of Hxt1-GFP and Hxt3-GFP in a *rod1Δ* context, contrary to Flag-tagged Rod1 (Flag) which functionally behaved like the wild-type protein (**S4A** and **S4B Fig**). We then re-evaluated Rod1 post-translational modifications using this functional construct. As for Rod1-3HA [32], we observed drastic changes in Rod1 mobility on gel after 2DG treatment (**Fig 2A**). In glucose-

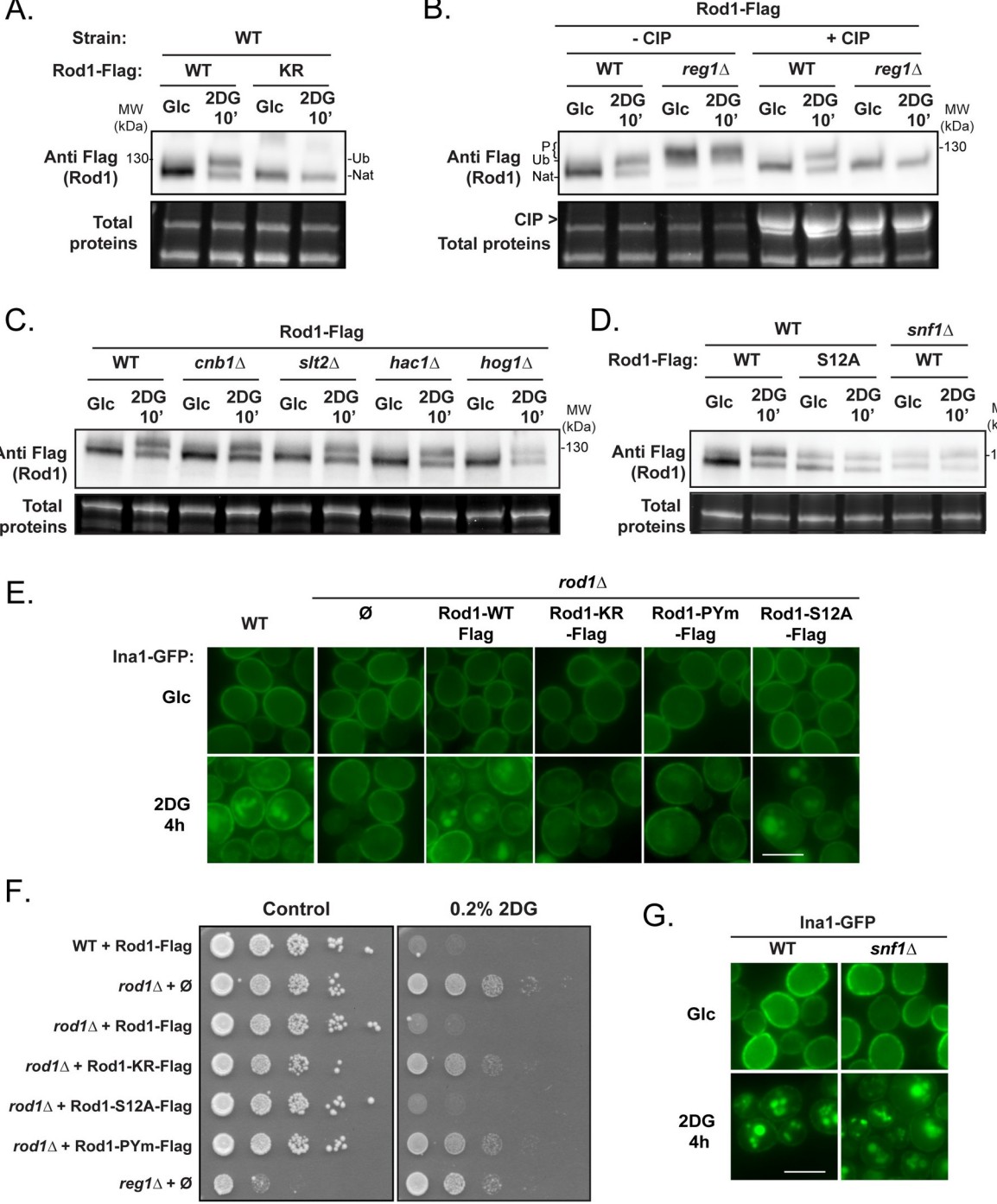

**Fig 2. Following 2DG treatment, Rod1 is dephosphorylated in a PP1-dependent manner.** (A) Total protein extracts of WT cells expressing either Rod1-Flag or Rod1-KR-Flag were prepared before and after 2DG addition for 10', and immunoblotted using an anti-Flag antibody. Ub: ubiquitylated Rod1, Nat: native Rod1. **(B)** Total protein extracts of WT or *reg1Δ* cells expressing Rod1-Flag were prepared before and after 2DG addition for 10', and immunoblotted using an anti-Flag antibody. Samples were dephosphorylated by CIP where indicated. Ub: ubiquitylated Rod1, Nat: native Rod1, P: phosphorylated Rod1. **(C)** Total protein extracts of WT, *cnb1Δ*, *slt2Δ*, *hac1Δ*, or *hog1Δ* expressing Rod1-Flag were prepared before and after 2DG addition for 10', and immunoblotted using an anti-Flag antibody. **(D)** Total protein extract of WT or *snf1Δ* cells expressing Rod1-Flag or Rod1-S12A-Flag were prepared before and after 2DG addition for 10' and immunoblotted using an anti-Flag antibody. **(E)** WT or *rod1Δ* cells expressing the various *ROD1*-Flag constructs were grown in a glucose-containing medium and observed by fluorescence microscopy before and after 2DG treatment for 4h. Scale bar, 5 μm. **(F)** Serial dilutions of cultures of the indicated strains were spotted on SC medium or SC + 0.2% 2DG medium and grown for 4 days at 30˚C. **(G)** WT or *snf1Δ* cells expressing Ina1-GFP were grown in a glucose-containing medium and observed by fluorescence microscopy before and after 2DG treatment for 4h. Scale bar, 5 μm.

grown cells, Rod1-Flag migrated as a diffuse band resulting from a mild phosphorylation, as determined by phosphatase treatment, and similarly to previously published results using Rod1-3HA [28,32] (**Fig 2A**). This pattern gave rise to two discrete bands upon 2DG exposure, with the upper band corresponding to ubiquitylated Rod1, as shown by its disappearance when mutating the known ubiquitylated lysine(s) (K235, K245, K264, K267) on Rod1 as described previously [28,32]. The lower band corresponded to a dephosphorylated species of Rod1, as demonstrated by the fact that it migrated similarly as Rod1-Flag from phosphatase-treated extracts (**Fig 2B**). This fits with previous results obtained in the context of glucose-induced endocytosis, in which Rod1 ubiquitylation was preceded by its dephosphorylation [28,29]. As expected, Rod1 was constitutively phosphorylated in the PP1 phosphatase mutant *reg1Δ* and this phosphorylation was no longer affected by 2DG treatment (**Fig 2B**). In contrast, mutations in the described Rod1 phosphatase, calcineurin (*cnb1Δ*) [36, 37] or in various signaling pathways whose activation is triggered by 2DG treatment [20] (MAPK = *slt2Δ*, *hog1Δ*; UPR: *hac1Δ*) had no impact (**Fig 2C**). This suggested that PP1 activity is required for the 2DG-induced dephosphorylation of Rod1. Moreover, deletion of *SNF1* or mutation of 12 serine residues within potential Snf1 consensus sequences (see Material and Methods) [36] led to a profile comparable to that obtained in 2DG-treated cells, suggesting that lack of Rod1 phosphorylation mimics 2DG-treatment (**Fig 2D**).

Having confirmed how Rod1-Flag is post-translationally modified in response to 2DG, we assessed the functional consequences abrogating these modifications on the function of Rod1 in endocytosis. Instead of focusing on Hxt1 or Hxt3, which are glucose-regulated [32], we used Ina1-GFP as a model cargo (see **Figs 1D** and **S5**). Ina1 is a protein of the SUR7/PalI family, with carries 3 predicted transmembrane domains and localizes to the plasma membrane [38]. Mutation of the ubiquitylation site(s) within Rod1 resulted in a slowdown of Ina1 endocytosis, whereas mutation of the PPxY motifs preventing the interaction with Rsp5 [28] completely abolished it (**Fig 2E**). This was confirmed by data obtained on the endocytosis of Hxt1-GFP and Hxt3-GFP (**S6 Fig**). Growth assays demonstrate that Rod1 activity in endocytosis correlated with 2DG sensitivity (**Fig 2F**). Interestingly, despite the fact that Rod1-S12A-Flag appeared constitutively dephosphorylated/ubiquitylated (see **Fig 2D**), Ina1-GFP endocytosis still relied on 2DG addition, suggesting yet another layer of regulation. Accordingly, Ina1 endocytosis was still regulated by 2DG in the *snf1Δ* mutant (**Fig 2G**), and so were other Rod1 cargoes (**S7 Fig**). Thus, Rod1 dephosphorylation (or lack of phosphorylation) is not sufficient to trigger endocytosis, suggesting yet an additional level of regulation.

## 2DG promotes the PP1-dependent dephosphorylation of Snf1 and several of its targets

The fact that 2DG triggers a PP1-dependent dephosphorylation of Rod1 contrasted with previous results showing that 2DG induces a mild phosphorylation (activation) of the PP1 substrate Snf1 within 2h of treatment [16], similar to observations in mammalian cells [3,39]. To explain this discrepancy, we reexamined the effects of 2DG on Snf1 signaling. For this, we followed Snf1 phosphorylation after 2DG treatment using antibodies directed against the activated (phosphorylated) form of human AMPKα, which cross-reacts with yeast Snf1 [40]. As a control for total Snf1, anti-polyhistidine antibodies can be used because Snf1 contains a stretch of 13 histidine residues that can be used for its detection [40]. However, the signals were sometimes very weak and so we completed our results with an endogenously GFP-tagged version of Snf1. We found this version to be fully functional (**S8 Fig**), as follows. Briefly, (i) the Snf1-GFP strain was able to use sucrose as a C source (**S8A Fig**), (ii) Mig1-Flag phosphorylation was similar in the Snf1-GFP strain as it was in the WT upon varying glucose concentrations, (iii)

Snf1-GFP phosphorylation increased at low glucose concentrations, which was counteracted by glucose addition, and finally (iv) Snf1-GFP phosphorylation increased in the *reg1Δ* mutant (**S8B Fig**). These data validate Snf1-GFP as a suitable epitope-tagged version of Snf1.

After 10 min 2DG treatment, we never observed Snf1 activation, even at higher 2DG concentration (**Fig 3A**). On the contrary, 2DG consistently decreased Snf1 phosphorylation within minutes after treatment (**Fig 3B and 3C**). A kinetic analysis of phospho-Snf1 response to 2DG confirmed a drop in Snf1 phosphorylation that was followed by an increase at later time points (**Fig 3C**). Snf1 dephosphorylation by 2DG required the PP1 subunit Reg1, thus mimicking the effect that glucose has on glucose-starved cells. Similar observations were made for the Snf1 substrate Rod1-Flag (**Fig 3C**). Altogether, we conclude that 2DG triggers an immediate and transient PP1-dependent dephosphorylation of Snf1, consistent with our (**Fig 2**) and others' [32] observations on Rod1.

This was further confirmed by studying another Snf1 substrate, the transcriptional repressor Mig1 [41,42] which was also dephosphorylated after 2DG treatment in a PP1-dependent manner (**Fig 3D**). Again, this mimicked the situation described when glucose-starved cells are treated with glucose [43]. Thus, 2DG triggers both an immediate and transitory PP1-dependent inhibition of Snf1, and the dephosphorylation of Snf1 substrates such as Mig1 or Rod1.

## Snf1 inactivation by 2DG requires 2DG phosphorylation by Hxk2

We then investigated the origin of the signal triggering the PP1-dependent dephosphorylation of substrates. 2DG is taken up by hexose transporters and is then phosphorylated into 2DG6P [44], which interferes with several metabolic pathways. Hxk2 is the main hexokinase isoform expressed in glucose-grown yeast cells [45] and we studied its contribution in 2DG phosphorylation by measuring 2DG6P appearance after 2DG treatment using an enzyme-based assay [46]. After 15 min exposure to 0.2% 2DG, 2DG6P was readily detectable in WT cells (**Fig 4A**). In *hxk2Δ*, 2DG6P only reached 25% of the WT value (**Fig 4A**), suggesting that Hxk2 is the main 2DG-phosphorylating enzyme.

2DG phosphorylation by hexokinase into the dead-end metabolite 2DG6P requires ATP and causes a strong ATP depletion [47, 48]. This can be visualized within minutes after 2DG addition in WT cells using an optimized, pH-insensitive version of the AT1.03 ATP FRET biosensor [49] (**Fig 4B**). The decrease in FRET signal occurred only upon 2DG treatment and revealed changes in ATP content as demonstrated using a mutant version of the FRET sensor which does not bind ATP [49] (**Fig 4B**). We found that comparatively to the WT, the drop in FRET was lower in *hxk2Δ* cells, again suggesting that Hxk2 is key for 2DG phosphorylation (**Fig 4C**).

In line with these findings, we found that neither Snf1 nor Rod1 were dephosphorylated in response to 2DG in the *hxk2Δ* mutant (**Fig 4D**). Similarly, 2DG no longer triggered Ina1-GFP endocytosis in this strain (**Fig 4E**), nor that of Hxt1 and Hxt3 (**S9A Fig**). Overall, our data suggest that either Hxk2 itself or 2DG phosphorylation is required for the 2DG-mediated activation of PP1. Noteworthy, overexpression of the 2DG6P phosphatase Dog2, but not that of its catalytic-dead mutant (Dog2-DDAA), also abolished Ina1-GFP endocytosis (**Fig 4F**) and that of Hxt1 and Hxt3 (**S9B Fig**), suggesting that 2DG-6-P levels directly or indirectly control PP1 activity. Altogether, we propose that 2DG phosphorylation by Hxk2 triggers a PP1-dependent Snf1 dephosphorylation, Rod1 dephosphorylation and endocytosis.

## Stabilization of glucose transporters at the plasma membrane confers 2DG resistance to the *rod1Δ* mutant

So far, our data provide mechanistic insights as to how 2DG triggers endocytosis through AMPK inhibition and Rod1 activation. *ROD1* deletion leads to 2DG resistance [14], but the

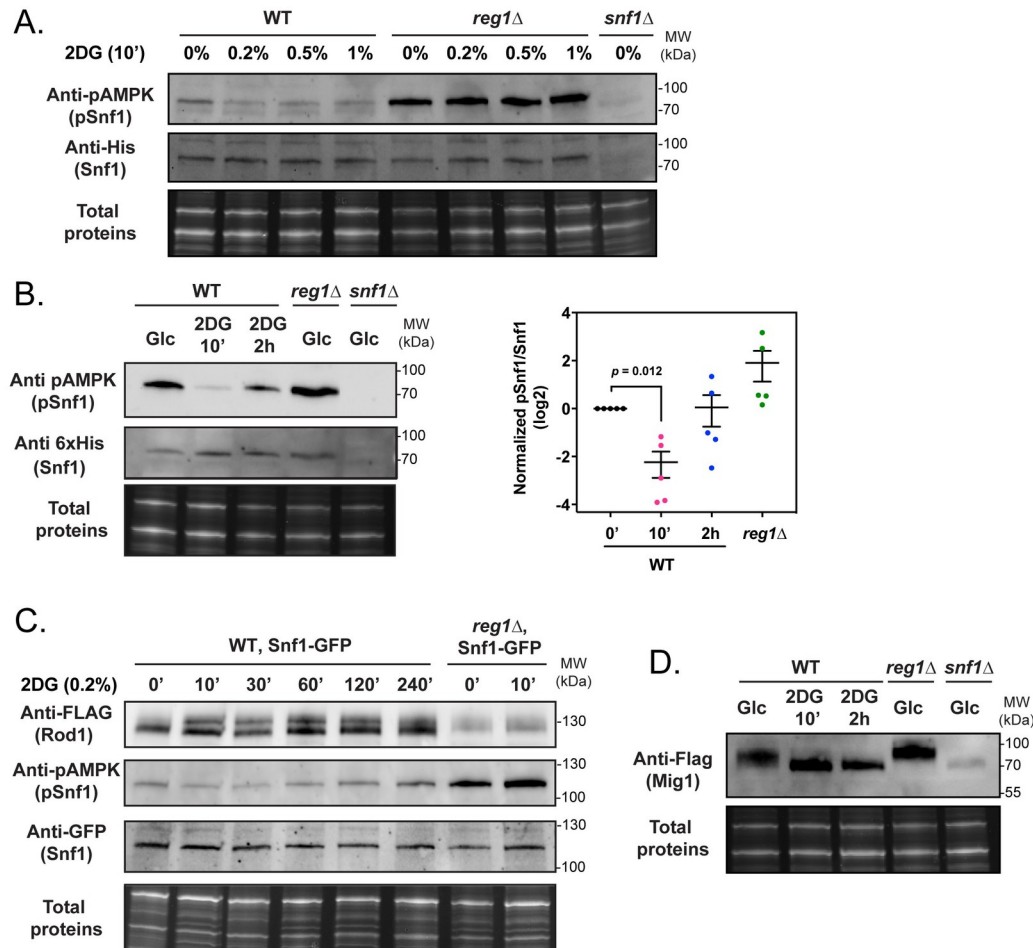

**Fig 3. 2DG causes a fast and temporary dephosphorylation of Snf1 and its substrates through PP1 activation. (A)** Total protein extracts of WT, *reg1Δ* and *snf1Δ* cells were prepared before and after 2DG addition at the indicated concentrations for 10 min and immunoblotted using anti-phospho-AMPK and anti-polyHistidine antibodies. **(B)** *Left*, Total protein extracts of WT, *reg1Δ* and *snf1Δ* cells were prepared before and after 2DG addition for the indicated time and immunoblotted using anti-phospho-AMPK and anti-polyHistidine antibodies. *Right*, quantification of the signals (adjusted *p*-value is indicated, one-way ANOVA). **(C)** Total protein extracts of WT and *reg1Δ* cells expressing Snf1-GFP and Rod1-Flag were prepared before and after 2DG addition for the indicated time and immunoblotted using anti-Flag, anti-phospho-AMPK and anti-GFP antibodies. **(D)** Total protein extracts of WT, *reg1Δ* and *snf1Δ* cells expressing Mig1-Flag were prepared before or after 2DG addition for 10' and 2h and immunoblotted using anti-Flag antibodies.

underlying mechanism remains unclear. We previously described that a frequent strategy leading to 2DG resistance involves an increased expression of the Dog2 phosphatase at the transcriptional level [20]. However, using a *pDOG2*:LacZ reporter, we found that this was not the case in the *rod1Δ* mutant, suggesting a distinct mechanism (**Fig 5A**).

The 2DG resistance of the *rod1Δ* strain is exacerbated upon the further deletion of its paralogue *ROG3* [32], and other data indicate that 2DG resistance correlates with Rod1 activity in endocytosis [32] (see also **Fig 2F**). Since Rod1 has a central role in plasma membrane protein remodeling by 2DG (**Figs 1D** and **S2**), we first hypothesized that the 2DG-mediated activation of Rod1 might trigger excessive endocytosis that would be deleterious for the cell, and could be counteracted by *ROD1* deletion. We postulated that 2DG resistance originates from the stabilization of one or several Rod1-regulated cargoes at the plasma membrane. We initially focused on the low affinity hexose transporters, Hxt1 and Hxt3, for the following reasons. First, the

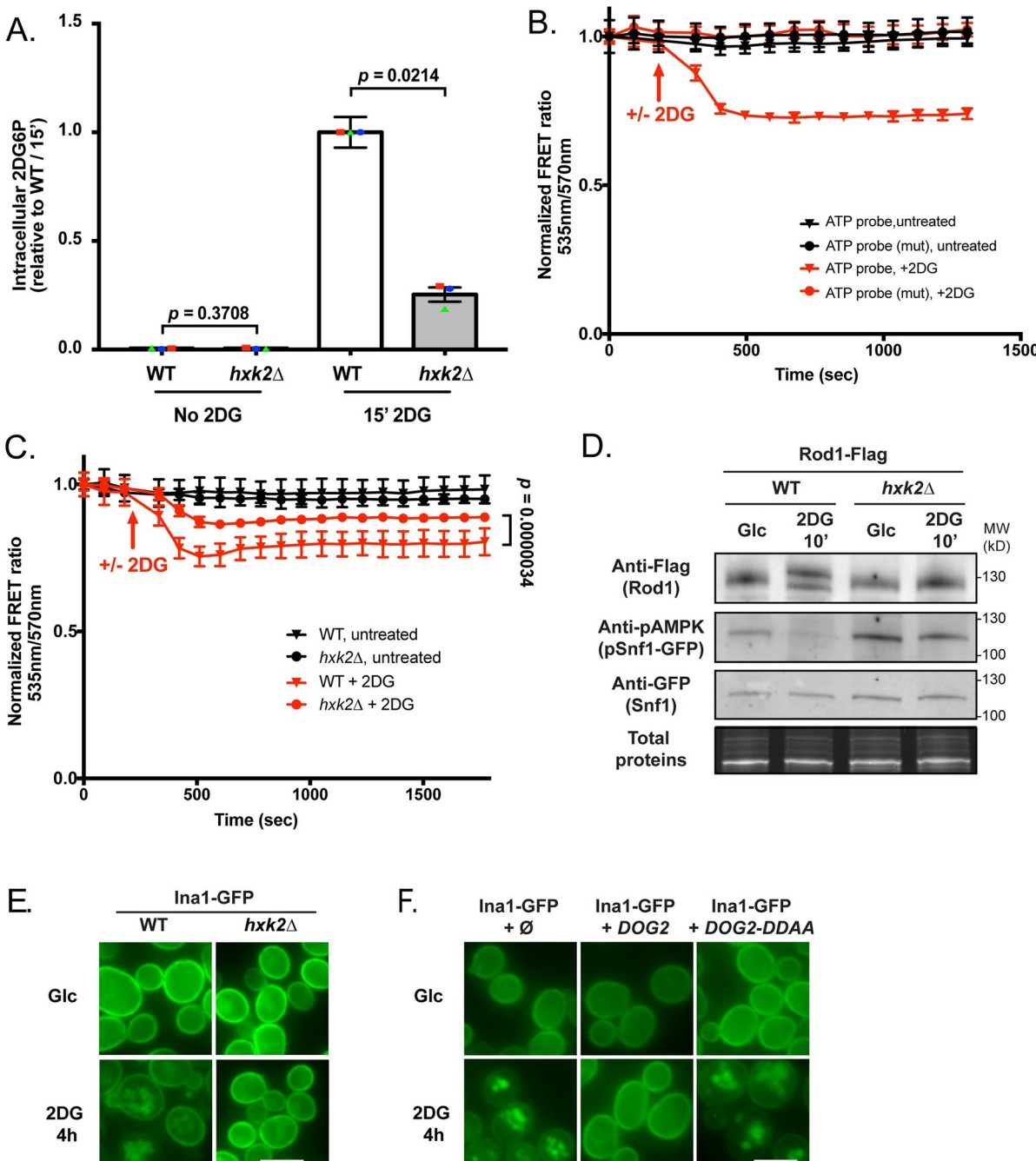

**Fig 4. Hexokinase 2 (Hxk2) is the main 2DG-phosphorylating enzyme *in vivo*.** (**A**) Intracellular 2DG6P was assayed enzymatically (see Methods) in WT and *hxk2Δ* cells grown overnight in a glucose-containing medium and treated or not for 15 min with 0.2% 2DG. Values are normalized to the value of the WT / 15 min (n = 3 independent experiments ± SEM, *p*value indicated, paired *t*-test) (**B**) 2DG causes a decrease in ATP content as visualized using a FRET ATP biosensor. WT cells expressing the ATP biosensor (WT or mutated version) were grown overnight in a glucose-containing medium and treated with 0.2% 2DG (arrow). The FRET ratio (535/570 nm) was measured over time in a plate reader (see Methods) and is represented as normalized to the t0 value. (*n* = 3 independent experiments ± SEM). (**C**) ATP levels were measured as in (B) in WT and *hxk2Δ* cells in response to 2DG. The FRET ratio (535/570 nm) is represented as normalized to the t0 value (n = 4 independent experiments ± SEM). A paired *t*-test was used to compare WT + 2DG vs *hxk2Δ* + 2DG (*p*-value indicated). (**D**) Total protein extracts of WT and *hxk2Δ* expressing Snf1-GFP and Rod1-Flag were prepared before and after 10' 2DG treatment and immunoblotted using anti-Flag, anti-phospho-AMPK and anti-GFP antibodies. (**E**) WT or *hxk2Δ* cells expressing Ina1-GFP were grown in a glucose-containing medium and observed by fluorescence microscopy before and after 2DG treatment for 4h. Scale bar, 5 μm. (**F**) WT cells expressing Ina1-GFP and transformed with an empty plasmid, or with plasmids allowing the overexpression of *DOG2* or its catalytic mutant *DOG2-DDAA* were grown in a glucose-containing medium and observed by fluorescence microscopy before and after 2DG treatment for 4h. Scale bar, 5 μm.

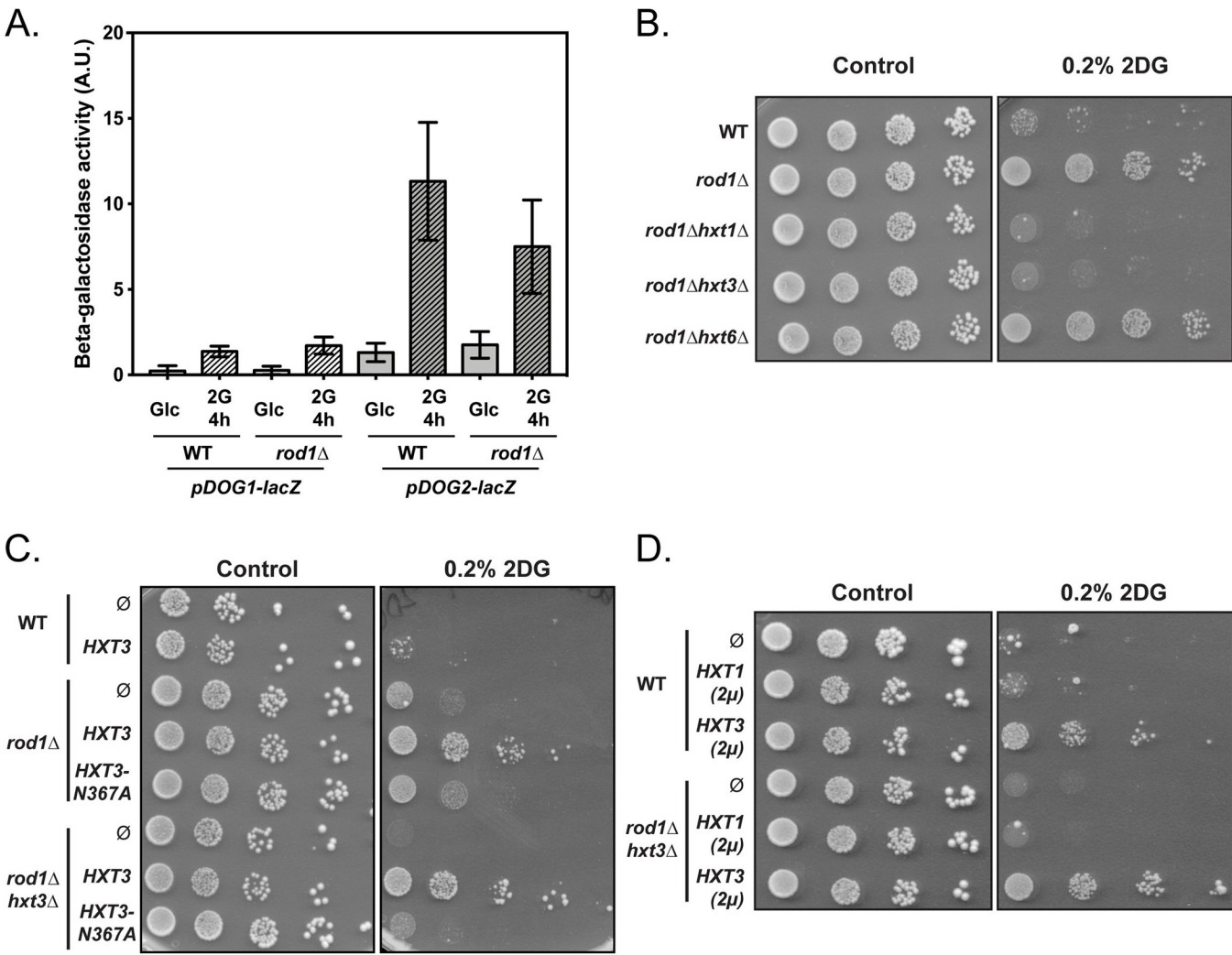

**Fig 5. Low affinity glucose transporters are required for the resistance of the *rod1Δ* mutant to 2DG.** (**A**) Beta-galactosidase assays of WT and *rod1Δ* cells expressing *LacZ* under the control of the p*DOG1* or p*DOG2* promoters, before and after 2DG treatments for 4h (+- SEM, n = 3 independent experiments, *t*-test). A.U., arbitrary units. (**B-D**) Serial dilutions of cultures of the indicated strains were spotted on SC medium or SC + 0.2% 2DG medium and grown for 4 days at 30°C.

hypersensitivity of the *snf1Δ* mutant to 2DG is rescued by overexpression of Hxt1 and Hxt3 through an unknown mechanism [32]. Second, duplication of a chromosomal region containing hexose transporter genes (*HXT3*, *HXT6* and *HXT7*) confers 2DG-resistance in a dominant manner [22]. Since these transporters are not endocytosed in the *rod1Δ* mutant (**Figs 1C** and **S2**) [32], we thus considered that their stabilization could contribute to 2DG resistance. Indeed, deletion of either *HXT3* or *HXT1* abolished the 2DG resistance of the *rod1Δ* mutant strain (**Fig 5B**), showing that both transporters are required for this process. In contrast, deletion of the high-affinity hexose transporter gene *HXT6* did not cause the same phenotype (**Fig 5B**). Although, it seems that only low-affinity hexose transporters promote 2DG resistance, this conclusion may be biased by the fact that low- and high-affinity hexose transporters are expressed to a lower level, the former contributing more to glucose uptake than the latter in glucose-rich conditions [50].

The growth defect of the obtained *rod1Δ hxt3Δ* strain on 2DG was restored by expressing a GFP-tagged version of Hxt3 driven by its own promoter on a low-copy (centromeric) plasmid (**Fig 5C**). This mild overexpression of Hxt3 also further increased 2DG resistance of the already resistant *rod1Δ* strain but had no effect on WT cells. Only a stronger expression of Hxt3 from a multicopy (2μ) plasmid conferred resistance to WT cells (**Fig 5D**), whereas that of Hxt1 had little or no effect as found previously [32]. Hxt1 and Hxt3 may have different functions regarding 2DG resistance, since Hxt1 overexpression did not compensate for the loss of Hxt3 in the *rod1Δ hxt3Δ* background (**Fig 5D**). Overall, we conclude that a high expression of glucose transporters generally correlates with 2DG resistance.

Structure/function studies of transporters (reviewed in [51]) identified residues critical for hexose binding and translocation. In particular, residue N317 of the human glucose transporter GLUT1 binds glucose [52,53] and mutation of the corresponding residue in yeast transporters alters substrate specificity and transport [54–56]. Particularly, mutation of this residue in Hxt1 (N370A) [57] and other yeast hexose transporters [54,58] abolished glucose uptake. We mutated the corresponding residue in Hxt3 (N367A) and found that its overexpression no longer rescued growth on 2DG, further linking glucose transporter function with 2DG resistance (**Fig 5C**).

## Stabilization of glucose transporters at the plasma membrane facilitates glucose uptake and 2DG export

The fact that glucose transport correlates with 2DG resistance was counterintuitive because 2DG enters into cells through glucose transporters [44], thus their stabilization at the cell surface may increase 2DG uptake. Indeed, deleting both *HXT1* and *HXT3* causes partial resistance to 2DG (**Fig 6A**), indicating that 2DG is mostly transported into cells through these transporters.

We considered two hypotheses to explain how glucose transporter accumulation at the plasma membrane results in 2DG resistance. First, by depleting glucose transporters from the plasma membrane, 2DG-induced endocytosis could lower glucose intake at a time where glycolysis is already impacted, resulting in energy shortage as previously proposed [32]. Preventing endocytosis may therefore attenuate the glycolysis blockade by maintaining glucose uptake. Alternatively, transporters present at the cell surface may instead favor 2DG detoxification and release into the medium, as previously observed [44] since the directionality of transport (import or export) through hexose transporters is dictated by the concentration gradient.

We first tested whether *rod1Δ* cells could detoxify 2DG6P more efficiently than WT cells. Cells were treated for 2h with 2DG, and 2DG6P was generated at comparable levels in both strains (**Fig 6B**). Interestingly, in WT cells, the concentration of 2DG6P after 2h 2DG treatment was comparable to that observed after 15 min (**Fig 6C**), showing that saturation is reached very early after treatment. Cells were then transferred to a 2DG-free medium for 30 min and 2DG6P was assayed again. In both strains, 2DG6P concentration decreased during these 30 min, revealing they could detoxify 2DG6P (**Fig 6B**). However, the 2DG6P content was much lower in *rod1Δ* than in WT cells, suggesting that 2DG6P is better detoxified in this strain. Therefore, the endocytosis blockade caused by the deletion of *ROD1* allows a better detoxification of 2DG. To address whether this involved Hxt3, we repeated this experiment in a *rod1Δ hxt3Δ* context (**Fig 6D**). We found that this restored 2DG6P content to comparable levels as those of the WT strain, whereas the deletion of *HXT3* alone had no significant effect on 2DG6P accumulation or disappearance. Therefore, Hxt3 is required for the increased detoxification of 2DG6P observed in the *rod1Δ* cells.

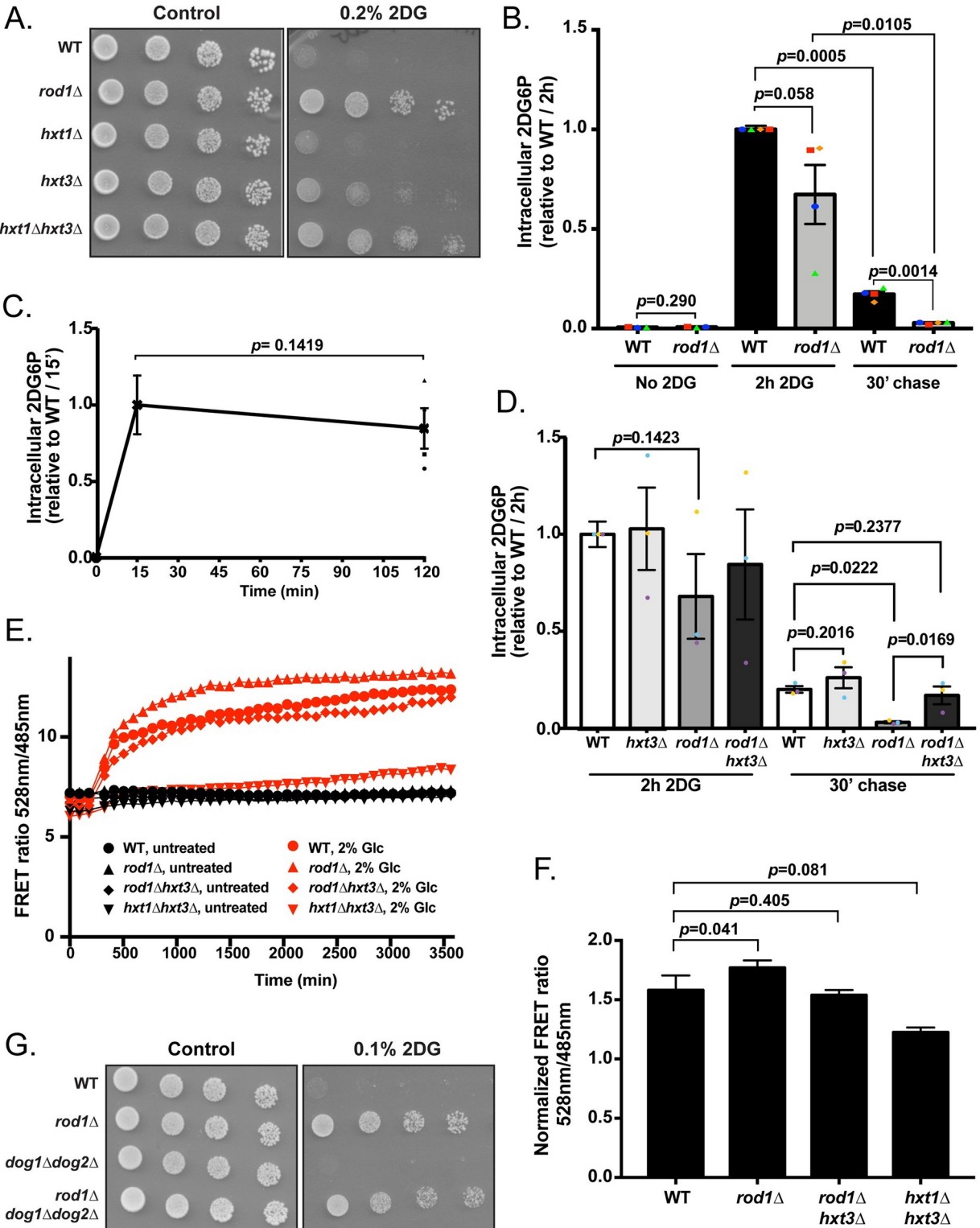

**Fig 6. The maintenance of glucose transporters in the *rod1Δ* mutant promotes 2DG resistance by increasing glucose uptake and limiting 2DG toxicity.** (**A**) Serial dilutions of cultures of the indicated strains were spotted on SC medium or SC + 0.2% 2DG medium and grown for 4 days at 30˚C. (**B**) Intracellular 2DG6P was assayed enzymatically in WT and *rod1Δ* cells grown overnight in a glucose-containing medium, treated for 2h with 0.2% 2DG, and then transferred into a 2DG-free glucose-containing medium ("chase"). Values are normalized to the value of the WT / 2h (n = 4 independent experiments ± SEM, paired *t*-test, *p*-value indicated). (**C**) 2DG6P content was assayed from WT cells as in (**B**)

after 15 min and 2h treatment with 0.2% 2DG n = 4 independent experiments ± SEM, paired *t*-test, *p*-value indicated). (**D**) Intracellular 2DG6P was assayed enzymatically in WT, *hxt3Δ*, *rod1Δ* and *rod1Δ hxt3Δ* cells as in (**B**). Values are normalized to the value of the WT / 2h (n = 3 independent experiments ± SEM, paired *t*-test, *p*-value indicated). (**E**) Intracellular glucose measurement using a FRET-based glucose biosensor (representative experiment). Cells were grown overnight in glucose medium (exponential phase), treated for 4h with 0.2% 2DG, and transferred in a glucose-free buffer. Fluorescence at 485 and 528 nm was measured every 90 sec in a plate reader before and after glucose addition (2%) (see Methods) and the FRET ratio (485/528 nm) is indicated over time. (**F**) Intracellular glucose was evaluated as in (**D**) at 2000 sec across n = 5 independent experiments (± SEM, paired *t*-test) and is represented as normalized to the "before glucose" value. (**G**) Serial dilutions of cultures of the indicated strains were spotted on SC medium or SC + 0,1% 2DG medium and grown for 4 days at 30˚C.

We also tested whether hexose transporters maintenance at the membrane impacts on glucose uptake, by using an intracellular glucose FRET sensor based on the bacterial glucose/galactose-binding transport protein MglB [59]. Previous work established that 2DG does not bind significantly to this probe, allowing its use to measure glucose content in our setting [60]. We assessed intracellular glucose following glucose exposure after a 4h-treatment with 2DG, and found that glucose accumulated slightly more in *rod1Δ* than in WT cells (**Fig 6E and 6F**). This depended on the presence of the glucose transporter Hxt3, since the additional deletion of *HXT3* in the context of *rod1Δ* restored glucose uptake to WT levels (**Fig 6E and 6F**) and 2DG sensitivity (**Fig 5B–5D**). Finally, deletion of both *HXT1* and *HXT3* strongly impaired glucose uptake, as expected. Thus, *ROD1* deletion could facilitate cell survival independently of 2DG detoxification by allowing a sustained glucose uptake. This was further illustrated by the fact that deletion of *ROD1* conferred 2DG resistance to a strain lacking both *DOG1* and *DOG2*, that are central to 2DG detoxification (**Fig 6G**). Therefore, the lack of endocytosis of hexose transporters protects cells from 2DG toxicity by several mechanisms.

## Discussion

Several strategies are known to modulate 2DG sensitivity/resistance in yeast [3]. The most common mechanism is through the aberrant activation of the AMPK orthologue, Snf1, either by gain-of-function mutation in constituents of the AMPK heterotrimeric complex, or by loss-of-function mutations of its inhibitory phosphatase (PP1: Glc7/Reg1) [3,21]. How AMPK hyperactivation contributes to 2DG resistance is not fully understood, but we previously found that this partially occurs through the expression of the detoxification enzyme Dog2 [20]. Here, we provide important insights into the cellular effects that 2DG has on AMPK signaling, how this impacts Rod1 activity and subsequently, on endocytosis, and how endocytosis interference leads to 2DG resistance through the control of glucose transport activity.

We first report that 2DG acts as a strong endocytosis signal, leading to the endocytosis of many plasma membrane proteins. These include a GPCR (Ste2), various classes of transporters, such as permeases, solute carriers such as multidrug transporters of the major facilitator superfamily (MFS) or ATP-binding cassette (ABC) types. This massive 2DG-induced endocytosis of membrane proteins is reminiscent of that observed in other situations of metabolic stresses such as acute glucose or nitrogen starvation [61–63], or upon treatment with other drugs, stressors or heat shock [26,64–69]. Because 2DG is an artificial molecule, the observed endocytosis is unlikely to originate from an adaptation strategy but rather informs us that 2DG-induced metabolic blockade triggers endocytosis through mechanisms that we further explored.

First, our work highlights the major role played by the arrestin Rod1 in 2DG-induced endocytosis. Rod1 participates in the endocytosis of the glucose transporters Hxt1 and Hxt3 upon 2DG treatment [32], and we extend these findings to many other membrane proteins. Using a tagged but fully functional protein, we confirm previous findings [32] that 2DG treatment strongly affects Rod1 post-translational modifications (phosphorylation and ubiquitylation),

reminiscent of past studies focusing on the regulation of Rod1 by glucose availability [28,29,31,36]. Phosphorylation is a frequent mechanism for arrestin inhibition [28,65,70,71], perhaps by introducing negative charges that may hinder transporter interactions [63] [reviewed in 27]. Accordingly, Rod1 dephosphorylation and its subsequent ubiquitylation are required for its activity in endocytosis [28,29]. Arrestin ubiquitylation is often required for their function [28,70,72], and this may favor interaction with Rsp5 and thus the Rsp5-mediated ubiquitylation of transporters [73]. Overall, we propose that 2DG induces an aberrant activation of Rod1 which explains the increased endocytosis observed in response to 2DG.

Second, we gained insights into how 2DG triggers these changes on Rod1, given that the precise effect of 2DG on Rod1 regulatory kinase(s) or phosphatase(s) was unclear until now. 2DG is well known to trigger AMPK activation in mammalian cells even at short timepoints [3,39,74], and a similar but milder effect was reported for Snf1 in yeast after 2h treatment [16]. Yet, this apparent activation of Snf1 did not translate into an increased phosphorylation of its substrate Mig1 [16], and the reported dephosphorylation of Rod1 in these conditions was also not in line with an increased Snf1 activity. Rather, by focusing on shorter timepoints, we observed a sharp decrease in Snf1 phosphorylation shortly after 2DG addition. Thus, 2DG treatment rapidly inhibits yeast Snf1 activity, contrary to observations on mammalian AMPK. We also found that after this initial drop in phosphorylation, Snf1 activity slowly increases over time to reach back its initial activity, or even higher levels [16]. However, we expect that hours of 2DG treatment strongly perturb cellular physiology and it may be delicate to specifically pinpoint the mechanisms leading to Snf1 activation in this context.

Beyond the fact that it involves PP1, the mechanism by which Snf1 dephosphorylation is stimulated by 2DG shortly after treatment remains to be established. However, several possibilities can be considered. First, through a derived metabolite or a metabolic imbalance caused by the glycolytic inhibition, 2DG could activate the Glc7/Reg1 PP1 subcomplex, for example by favoring its assembly or its recruitment onto substrates. A second and more likely possibility is derived from previous work on Snf1 dephosphorylation in response to glucose. Using analog-sensitive alleles of a Snf1-activating kinase or of Snf1 itself, it was shown that PP1 activity towards Snf1 is stimulated by glucose whereas PP1 activity towards Mig1 remains constant regardless of the glucose status [12]. Whether this is due to a better accessibility of Snf1, an increased recruitment of PP1, or a stimulation of PP1 activity, the same mechanism could be at stake in response to 2DG. The maintenance of PP1 activity towards Snf1 substrates (eg. Mig1) in a context where Snf1 is inactivated would be sufficient to explain why phosphorylation of Mig1 decreased in response to 2DG within a few minutes. This may also explain observations on Rod1 which also requires PP1 for its dephosphorylation in response to glucose [28,29] or 2DG (this study and [32]). The increased dephosphorylation of Snf1 in response to 2DG is reminiscent to that observed when glucose is added to cells deprived of glucose [75]. Thus, shortly after treatment and even in glucose-grown cells, 2DG treatment mimics glucose-replete conditions rather than starvation, and part of the early 2DG response involves a drop in Snf1 activity.

This inactivation of Snf1 by 2DG involves Hxk2, as 2DG was no longer able to trigger Snf1/Rod1 dephosphorylation or the endocytosis of Ina1, Hxt1 and Hxt3 in the *hxk2Δ* strain. Hxk2 may either have a unique function in 2DG toxicity beyond its enzymatic function, but we favor the hypothesis that Hxk2 mediates 2DG-induced signaling through its unique ability to generate 2DG6P. Despite the fact that 2DG6P can be generated by other sugar-phosphorylating enzymes (Hxk1 and glucokinase Glk1) *in vitro* [21], only *HXK2* deletion causes 2DG resistance [14,16,20,21]. Moreover, our *in vivo* data suggest that Hxk2 is the most prominent 2DG-phosphorylating enzyme, since *hxk2Δ* cells have a lower 2DG6P content and 2DG has a lower impact on the ATP status in these cells. The lack of effect of Hxk1 or Glk1 on 2DG toxicity

could be explained by a lower expression level or other *in vivo* regulatory mechanisms that would prevent an efficient 2DG phosphorylation. Interestingly, both *HXK2* deletion and over-expression of the 2DG-6-P phosphatase Dog2, which converts 2DG-6-P back into 2DG, protect *snf1Δ* against 2DG hypersensitivity [16,20], so both may act through a decreased 2DG6P content. More generally, 2DG has virtually no effect in strains overexpressing Dog2, further suggesting that 2DG6P (or a derived metabolite) contributes to the dephosphorylation of Snf1 and its substrates. Further work should be aimed at understanding how the PP1/Snf1 balance is controlled by 2DG at the molecular level, which may help decipher the long-sought mechanism by which glucose-starved cells react to glucose addition [12,75–77].

After delving into the mechanisms by which 2DG triggers endocytosis through the modulation of glucose signaling pathways and Rod1 activity, we also obtained insights into how *ROD1* deletion results in 2DG resistance. The study of Rod1 point mutants indicate a correlation between endocytosis and 2DG resistance (this work, and [32]) and Hxt1 and Hxt3 stabilization at the plasma membrane was necessary for 2DG resistance, in line with the fact that their overexpression allows 2DG resistance [32]. Although Hxt1 and Hxt3 are the main transporters involved in 2DG uptake, their maintenance at the plasma membrane is required for long-term resistance through two possible pathways that possibly synergize (Fig 7). The first pathway involves 2DG detoxification. Indeed, 2DG triggers the induction of the Dog1 and Dog2 phosphatases which dephosphorylate 2DG6P into 2DG [20], the latter of which could presumably be exported once its cytosolic concentration is higher than that of the medium and provided that hexose transporters be present at the plasma membrane. This is in line with observations showing that (i) transport across the membrane is required to confer resistance, (ii) the same transport systems are used both for 2DG and glucose [44], and (iii) proteins of the GLUT/HXT family support bidirectional fluxes across the membrane (eg. GLUT2 in hepatocytes, reviewed in [78]). Accordingly, we found that *ROD1* deletion increases the ability to detoxify 2DG6P, and this depended on the presence of Hxt3. Thus, stabilizing glucose transporters at the plasma membrane might allow the release of 2DG back to the medium, but this possibility should be further explored in the future. The second pathway posits that transporter stabilization at the plasma membrane allows a sustained glucose import, and thus may help bypass the 2DG-induced glycolytic blockade. Over time, 2DG6P synthesis and accumulation should inhibit hexokinase and thus blocks glucose phosphorylation, which itself drives glucose import [79] (reviewed in [80]), but given the existence of detoxification mechanisms, the glycolytic blockade may be only temporary. Increasing transporter availability should in principle boost glucose import and help glycolysis recover, and accordingly, *rod1Δ* cells contain more glucose than WT cells when exposed to glucose after 2DG treatment.

In mammalian cells, 2DG causes a fast energy depletion and AMPK activation [39]. This is exemplified by the opposite effect that 2DG has on the endocytosis of glucose transporters. In hepatocytes, the 2DG-induced activation of AMPK triggers the phosphorylation of the arrestin-like protein TXNIP, resulting in its degradation and stabilization of the glucose transporter GLUT1 [74], a response which is consistent with energy depletion and starvation response. In contrast, 2DG transiently inhibits yeast Snf1, thus promoting Rod1-mediated glucose transporter endocytosis (this work and [32]). This somewhat paradoxical response likely reveals that in yeast, 2DG causes energy deprivation but is also likely sensed as excess glucose (causing a decreased Snf1/AMPK activity). The differences in 2DG signaling between these two model systems may originate from evolutionary differences regarding their ability to detect glucose as a signaling molecule, in addition to being a source of energy. Glucose is the preferred carbon source for *Saccharomyces cerevisiae* and is mostly used by fermentation, implying a high glycolytic flux to sustain proliferation. Yeast has evolved various sensing systems at the plasma membrane to detect extracellular glucose, that are wired into various

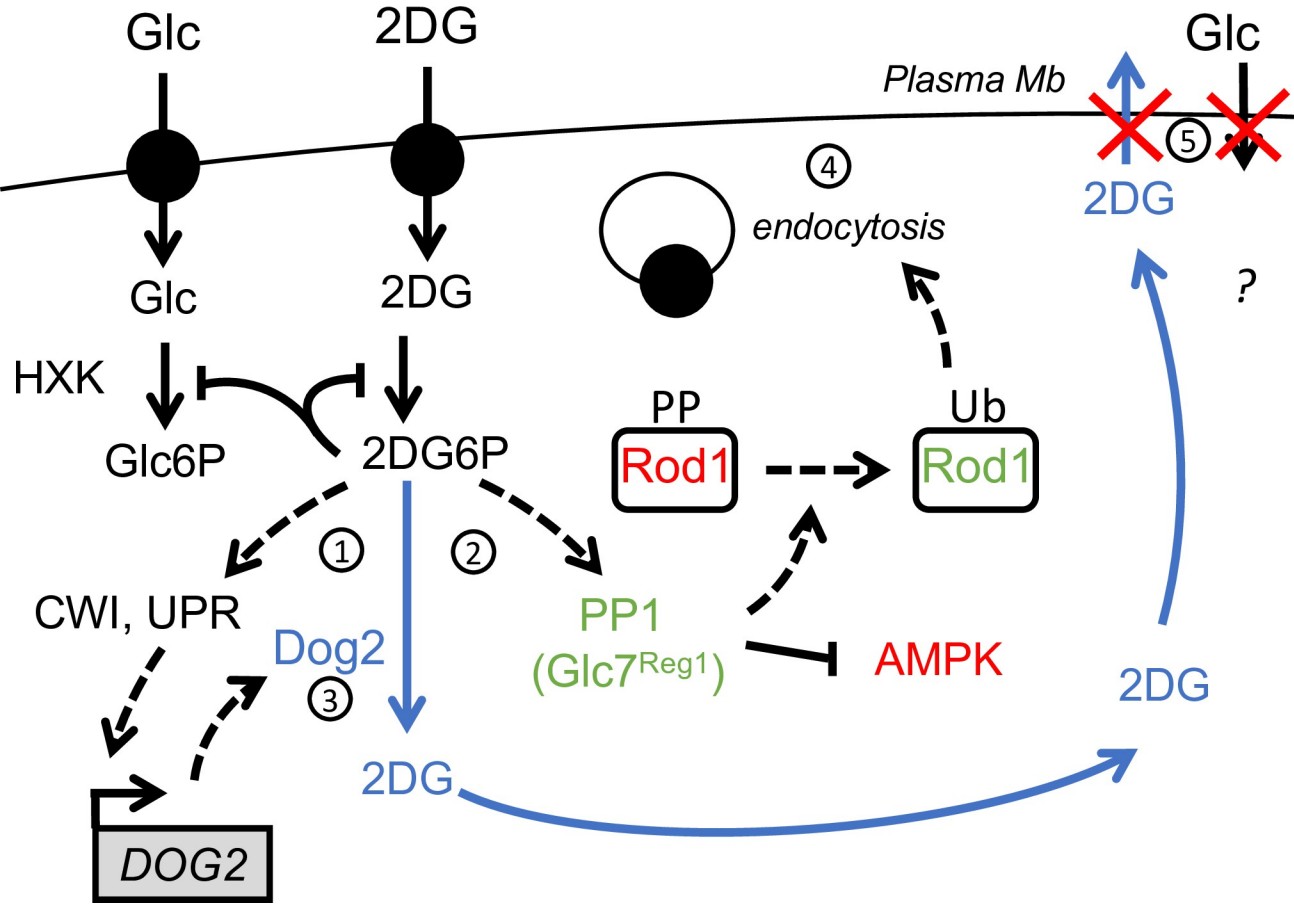

**Fig 7. Hypothetical model for a role of hexose transporters in 2DG detoxification.** Treatment of cells with 2DG triggers the accumulation of intracellular 2DG6P. This activates stress-signaling pathways such as cell-wall integrity signaling (CWI) or the unfolded protein response (UPR), converging onto *DOG2* expression (1) [20]. On the other hand, 2DG6P elicits a fast PP1 activation (2) which causes AMPK inhibition and, either directly or indirectly, Rod1 dephosphorylation. By the time Dog2 is induced and begins to dephosphorylate 2DG6P (3), Rod1 has already triggered the endocytosis of many plasma membrane proteins (4), including that of low affinity glucose transporters. This may prevent 2DG exit and detoxification but also affect cell survival by reducing glucose import (5).

signaling pathways [81]. It is likely that Hxk2 and/or glucose phosphorylation itself acts as a glucose sensor within the Snf1/AMPK pathway [82–84], and 2DG phosphorylation by Hxk2 may produce a glucose-replete signal, leading to Snf1 inactivation despite changes in the energy load. This is probably only true shortly after 2DG treatment, before 2DG inhibits other aspects of cellular physiology (such as protein glycosylation or structural sugar metabolism) leading to a complex response through stress-signaling pathway (such as UPR or MAPK-based signaling) [3,20]. In conclusion, the characterization of 2DG resistance strategies led once more to a better understanding of the cellular effects of 2DG and the diversity of mechanisms that can be deployed to bypass 2DG-induced metabolic inhibition.

## Materials and methods

### Yeast strain construction and growth conditions

All yeast strains used in this study derive from the *Saccharomyces cerevisiae* BY4741 or BY4742 background and are listed in S1 Table. Apart from the mutant strains obtained from the yeast deletion collection (Euroscarf) and the fluorescent GFP-tagged strains originating from the

yeast GFP clone collection [86], all yeast strains were constructed by transformation with the standard lithium acetate–polyethylene glycol protocol using homologous recombination and verified by polymerase chain reaction (PCR) on genomic DNA prepared with a lithium acetate (200 mM)/SDS (0.1%) method (104). Yeast cells were grown in YPD medium (2%) or in SC medium [containing yeast nitrogen base (1.7 g/liter; MP Biomedicals), ammonium sulfate (5 g/liter, Sigma-Aldrich), the appropriate drop-out amino acid preparations (MP Biomedicals), and 2% (w/v) glucose, unless otherwise indicated]. Precultures were incubated at 30˚C for 8 hours and diluted in the evening to 20-ml cultures to reach mid-log phase the next morning. 2DG (Sigma) was added to mid-log phase yeast cultures grown overnight to final concentrations of 0.2% (w/v) and incubated for the indicated times.

## Plasmid construction

All plasmids used in this study are listed in S2 Table.

Plasmid pSL553 (pRS313, $p_{ROD1}$:ROD1-NcoI-Flag) was generated by site-directed mutagenesis on pSL234 (pRS313-based, $p_{ROD1}$:ROD1-Flag, a kind gift of O. Vincent, CSIC, Madrid, Spain) to integrate a NcoI site between ROD1 and the Flag-tag, allowing to substitute the WT ORF of ROD1 by mutated versions at SalI-NcoI sites as follows. The inserts $p_{ROD1}$:ROD1-WT-Flag (pSL559), $p_{ROD1}$:ROD1-PYm-Flag (pSL560) $p_{ROD1}$:ROD1-S12A-Flag (pSL561) and $p_{ROD1}$:ROD1-KR-Flag (pSL563) were PCR amplified (oSL1662/oSL1663) using as templates pSL94 ($p_{ROD1}$:ROD1-3HA) [28], pSL119 ($p_{ROD1}$:ROD1-PYm-3HA) [28], pSL152 ($p_{ROD1}$:ROD1-S12A-3HA) (see below), and pSL147 ($p_{ROD1}$:ROD1-KR-3HA, carrying mutations K235R, K245R, K264R, K267R) [28] respectively. They were digested with SalI/NcoI and cloned at these sites into pSL553 ($p_{ROD1}$:ROD1-NcoI-Flag). Plasmid pSL152 (pRS415, $p_{ROD1}$-ROD1(S12A)-3HA) was made from a synthetic gene designed to mutate S125, S138, S315, S358, S447, S623, S641, S706, S720, S734, S781, and S821 residues into A (Eurofins), digested PacI/XmaI and cloned PacI/XmaI into pSL94 at the place of WT ROD1 (pRS415, $p_{ROD1}$:ROD1-3HA) [28].

To construct pSL589 (pUG35-based, $p_{HXT3}$-Hxt3-GFP), the HXT3 promoter (1kb) was first cloned into pUG35 after amplification by PCR on WT gDNA (oSL1739/oSL1740) at BamHI/SpeI sites to give pSL592. The HXT3-GFP ORF was PCR-amplified (oSL1770/oSL1771) from gDNA of the Hxt3-GFP strain (ySL1027) using primers overlapping with the pSL592 sequence and cloned into pSL592 by gap-repair, and later rescued in bacteria to obtain pSL589. The N367A mutation was introduced using 2 overlapping PCR products amplified from gDNA of the HXT3-GFP strain (ySL1027) (oSL1730/oSL682 and oSL1731/oSL1563) and cloned by gap-repair into pSL592 to give pSL591. Plasmid pSL599 (pRS426, $p_{HXT1}$:HXT1) was obtained by (1) amplifying the HXT1 gene (-515/+225) from WT genomic DNA (oSL435/oSL932) into a pCRII-blunt topo vector (pSL597), (2) cloning a BamHI/XhoI fragment of pSL597 into pRS425 at BamHI/XhoI sites. For plasmid pSL602 (pRS426, $p_{HXT1}$:HXT1), A PCR amplifying the HXT3 gene (and 1kb of promoter) was obtained from WT genomic DNA (oSL1739/oSL1734), digested BamhI/SpeI and cloned BamHI/SpeI in pRS426.

All constructs were verified by sequencing.

## Total protein extracts and immunoblotting

Yeast were always grown in SC medium. For each protein sample, 1.4 ml of culture was incubated with 140μl of 100% TCA for 10 min on ice to precipitate proteins, centrifuged at 16,000g at 4˚C for 10 min, and broken for 10 min with glass beads. Lysates were transferred to another 1.5-ml tube to remove glass beads and centrifuged for 5 min at 16,000g at 4˚C, supernatants were discarded, and protein pellets were resuspended in sample buffer [50 mM tris-HCl (pH

6.8), 100 mM dithiothreitol, 2% SDS, 0.1% bromophenol blue, and 10% glycerol, complemented with 50 mM tris-base (pH 8.8)] (50μl/initial OD). Protein samples were heated at 37°C for 5 min and 10 μl was loaded on SDS–polyacrylamide gel electrophoresis (PAGE) gels (4–20% Mini-PROTEAN TGX Stain-Free, Bio-Rad). After electrophoresis for 30 min at 200V, total proteins were visualized by in-gel fluorescence using a trihalo compound incorporated in SDS–PAGE gels (stain-free TGX gels, 4–20%; Bio-Rad) after 45 sec UV-induced photoactivation using a ChemiDoc MP imager (BioRad), serving as a loading control. Gels were transferred on nitrocellulose membranes for 60 min (100V) in a liquid transfer system (Bio-Rad). Membranes were blocked in Tris-buffered saline solution containing 0.5% Tween-20 (TBS-T) and 2% milk for 30 min and incubated for at least 2 hours with the corresponding primary antibodies. Membranes were washed 3x10 min in TBS-T and incubated for at least an hour with the corresponding secondary antibody (coupled with horseradish peroxidase). Membranes were then washed again 3x10 min in TBS-T and incubated with SuperSignal West Femto reagent (Thermo). Luminescence signals were acquired using a ChemiDoc MP (BioRad). Primary and secondary antibodies used in this study as well as their dilutions are listed in S3 Table.

## Treatment of protein extracts with calf intestinal phosphatase (CIP) or Endoglycosidase H (EndoH)

Calf Intestine Phosphatase (CIP, Roche ref 11097075001) was used to dephosphorylate proteins prior to electrophoresis. Samples were prepared using 20 μL of TCA protein extract (see "Protein extracts and immunoblotting", above), to which 2 μL of 10X CIP buffer supplied with the enzyme, as well as 0.6 μL of 1.5M Tris pH 8.8, 0.2 μL 0.1M $MgSO_4$, and 17.2 μL of $H_2O$ were added. The extract was split in 2 and 0.6 U CIP was added to one of the samples. Samples were incubated at 37°C for 3 h, and loaded on SDS-PAGE.

Endoglycosidase H (EndoH) (Promega V4871, 500 U/μL) was used to deglycosylate Ina1-GFP-containing samples. Samples were prepared using 10 μL of TCA protein extract (see "Protein extracts and immunoblotting", above) to which 2 μL of the supplied buffer, 1 μL of EndoH and 7 μL of water were added. A control sample without EndoH was also prepared. Samples were incubated for 2 h at 37°C and loaded on SDS-PAGE.

## Beta-galactosidase assays

Beta-galactosidase assays were performed using 1 ml of mid-log phase yeast cultures carrying the *pDOG1-LacZ* (pSL409) or *pDOG2-LacZ* plasmids (pSL410) [20], grown overnight to the mid-log phase in SC medium without uracil with 2% glucose and switched to the specified conditions. The OD(600 nm) of the culture was measured, and samples were taken and centrifuged at 16,000g at 4°C for 10 min. Cell pellets were snap-frozen in liquid nitrogen and resuspended in 800 μl of buffer Z (pH 7, 50 mM $NaH_2PO_4$, 45 mM $Na_2HPO_4$, 10 mM $MgSO_4$, 10 mM KCl, and 38 mM beta-mercaptoethanol). After addition of 160μl of ONPG (ortho-nitrophenyl-d-galactopyranoside, 4 mg/ml; Sigma-Aldrich), samples were incubated at 37°C. Enzymatic reactions were stopped in the linear phase (60-min incubation for *pDOG2-LacZ* and 120-min incubation for the *pDOG1-LacZ* plasmid, as per initial tests) [20] by addition of 400 μl of $Na_2CO_3$, and cell debris were discarded by centrifugation at 16,000g. The absorbance of clarified samples was measured with a spectrophotometer set at 420 nm.

Galactosidase activities (arbitrary units) were calculated using the formula 1000*[A420/(A600*t)], where A420 refers to the measured enzyme activity, A600 is the turbidity of the culture, and t is the incubation time. Each enzymatic assay was repeated independently at least three times.

## 2DG-6-phosphate assays

Cells were grown overnight in SC media until mid-log phase. 2DG was added into the culture to a final concentration of 0.2% for the indicated time. For the chase experiments, the cultures were centrifuged after 2 hours, washed with water and re-suspended in the same volume of SC medium without 2DG. For each time points, 1 OD equivalent of cells were taken, immediately placed on ice, centrifuged at 4˚C for 4' at 16,000g, washed once with cold PBS and lysed using Y-PER Yeast Protein Extraction Reagent (ThermoFisher Scientific) as previously described [85]. Briefly, 1 OD equivalent of cells was resuspended in 100μL of Y-PER buffer, incubated for 20' at 30˚C, and centrifuged for 4' at 16100g at room temperature. The supernatant (between 0.0125 and 0.0625 OD equivalents) was used for the assay and diluted in Y-PER when needed. The assay was performed using the 2DG uptake measurement kit (Cosmo Bio USA, Carlsbad, CA, USA;, Ref. CSR-OKP-PMG-K01H) following the manufacturer's protocol adapted for a total reaction volume of 100 μL. Absorbance at 420 nm was measured every minute for 1 hour on a SpectraMax M2 Microplate Reader (Molecular Devices). Measures taken between 20 and 35 minutes (linear range) were taken into account to compare the slopes. Each sample was assayed at least twice in each experiment and at least three independent experiments were performed. Mean values were calculated and were plotted with error bars representing SEM. Statistical significance was determined using a *t*-test for paired variables assuming a normal distribution of the values (unless otherwise stated) using GraphPad Prism7.

## Glucose FRET sensor experiments

Yeast transformed with the plasmid FLII12Pglu-700μδ6 (pSL590, [a gift from Wolf Frommer: Addgene #28002, 59] were grown overnight in SC-URA medium to reach an OD600nm of 0.3–0.7 in the morning. Cells were centrifuged at 3000g for 5 minutes and the cell pellets were resuspended in 20mM MES pH = 6 to a final OD of 0.6. 180μl of the cell suspension were distributed in 12 wells of a 96-well black PS microplate with flat bottom (Greiner). Fluorescence was measured with a plate reader (Spark, TECAN) set at 30˚C before the experiment was started.

A first measurement for 3 cycles was performed to establish a baseline (each cycle is around 90 seconds with 10 seconds shaking before each measurement of the fluorescence). Cells were excited at CFP wavelength excitation (428 nm) and the emission intensities at 485 nm (CFP) and 528 nm (Citrine) were collected. The cycle was then paused to inject 20μl of the glucose solutions using a multichannel pipette, and the plate rapidly reloaded in the reader to restart measurements for another 17 cycles. Data were collected and analyzed in an Excel spreadsheet and in Prism 7.0 (GraphPad). Emission intensity values of yeast transformed with an empty plasmid were subtracted from emissions values at 485 nm and 528 nm for each measurement at the different glucose concentrations. The emission intensity ratio for Citrine (528 nm) over CFP (485 nm) was calculated. All analyses were repeated (with three technical replicates) at least three times independently. Emission ratios were normalized to the average of the three initial ratio values before glucose addition.

## ATP FRET sensor experiments

Yeast transformed with the plasmid pDRF1-GW yAT1.03 (pSL608) [a gift from Bas Teusink: Addgene #132781, 49] were grown overnight in SC-URA medium to reach an OD600nm of 0.3–0.7 in the morning. FRET measurements were performed as above except cells were excited at 438 nm and the emission intensities at 483 nm (ymTq2Δ11) and 593nm (tdTomato)

were collected. A plasmid encoding a mutated sensor which does not bind ATP (pSL609) [a gift from Bas Teusink: Addgene #132782, 49] was used as a negative control.

## Drop tests

Yeast cells grown in liquid-rich or SC medium for at least 6 hours at 30˚C were adjusted to an optical density (600 nm) of 1. Serial 10-fold dilutions were prepared in 96-well plates and spotted on plates containing rich or SC medium containing 2% (w/v) agar and, when indicated, 2DG [0.05 or 0.2% (w/v)]. Plates were incubated at 30˚C for 3 to 4 days before scanning on a desktop scanner (Epson).

## Microscopy

Cells were mounted in synthetic complete medium with the appropriate carbon source and observed with a motorized Olympus BX-61 fluorescence microscope equipped with an Olympus PlanApo 100× oil-immersion objective (1.40 NA), a Spot 4.05 charge-coupled device camera and the MetaVue acquisition software (Molecular Devices; Sunnyvale, CA) and imaged at room temperature. GFP-tagged proteins were visualized using a Chroma GFP II filter (excitation 440–470 nm). mCherry-tagged proteins were visualized using an HcRed I filter (excitation 525–575 nm). Images were processed in ImageJ (NIH). Vacuolar staining was obtained by incubating cells grown in the desired condition with 100 μM CMAC (Life Technologies) for 10 min under agitation at 30˚C. Cells are then washed twice with water before observations with a fluorescence microscope equipped with a DAPI filter.

## Quantification of vacuolar fragmentation

Quantification of vacuolar fragmentation (S1B Fig) was performed with ImageJ. For each experiment and each strain, at least 30 cells were observed and the quantification was done blindly. Cells with more than 3 vacuoles were considered as cells with highly fragmented vacuoles.

## Quantification of Western-blots

Western blot quantifications were performed with ImageJ. The integrated density of proteins of interest was divided by the integrated density of total proteins. Results were normalized to the Glc ON condition of each protein. The ratio of undegraded protein is shown. The experiments were repeated three times. Datas were analyzed and processed with Prism7 software (GraphPad). The statistical test performed was a one-way analysis of variance (ANOVA) test.

## Statistical analysis

Mean values were calculated using a minimum of three independent measurements from three biological replicates and were plotted with error bars representing SEM. Statistical significance was determined using a *t*-test for paired variables assuming a normal distribution of the values using GraphPad Prism7.

## Supporting information

**S1 Table. Yeast strains used in this study.**
(DOCX)

**S2 Table. Plasmids used in this study.**
(DOCX)

**S3 Table. Antibodies used in this study.**
(DOCX)

**S1 Fig. 2DG treatment induces vacuolar fragmentation. A.** WT cells expressing Vph1 tagged with mCherry were grown overnight in glucose medium (exponential phase) and treated with 0.2% 2DG. Cells were collected and observed by fluorescence microscopy at the indicated times. Scale bar: 5 μm. **B.** Quantification of 2DG-induced vacuolar fragmentation (values ± SD, *n* = 3 independent experiments).
(TIF)

**S2 Fig. Most of the studied plasma membrane proteins are endocytosed in response to 2DG and this endocytosis is often Rod1-dependent.** WT or *rod1Δ* cells expressing the indicated membrane proteins tagged with GFP at their endogenous genomic locus were observed by fluorescence microscopy before and after treatment with 2DG for 4h. Scale bar, 5 μm.
(TIF)

**S3 Fig. Rog3 is not responsible for the observed Rod1-independent endocytosis of Lyp1-GFP in response to 2DG.** WT, *rod1Δ* and *rod1Δ rog3Δ* cells expressing Lyp1-GFP tagged at its endogenous genomic locus were observed by fluorescence microscopy before and after treatment with 2DG for 4h. Scale bar, 5 μm.
(TIF)

**S4 Fig. Expression of Rod1-3HA does not restore endocytosis nor sensitivity to 2DG in a *rod1Δ* context, contrary to Rod1-Flag.** (**A**) The indicated strains were observed by fluorescence microscopy after growth in a glucose-containing medium and after 2DG treatment for 4h. Scale bar, 5 μm. (**B**) Serial dilutions of cultures of the indicated mutants/plasmid combinations were spotted on the indicated media and grown for 3 days at 30°C.
(TIF)

**S5 Fig. Ina1-GFP is endocytosed in response to 2DG.** (**A**) WT or *rvs167Δ* strains expressing Ina1-GFP were observed by fluorescence microscopy after growth in a glucose-containing medium and after 2DG treatment for 4h. In addition, WT cells were incubated with CMAC to label the vacuole. Scale bar, 5 μm. (**B**) *Left*, Total protein extracts of WT cells expressing Ina1-GFP were prepared before and after 2h or 4h 2DG treatment and immunoblotted using anti-GFP antibodies. *Right*, the same extracts were treated with endoglycosidase H and immunoblotted using anti-GFP antibodies.
(TIF)

**S6 Fig. Endocytosis of Hxt1-GFP and Hxt3-GFP requires Rod1 interaction with Rsp5 and Rod1 ubiquitylation.** The indicated strains were observed by fluorescence microscopy after growth in a glucose-containing medium and after 2DG treatment for 4h. Scale bar, 5 μm.
(TIF)

**S7 Fig. *SNF1* deletion is not sufficient to block endocytosis in response to 2DG.** Pdr12-GFP, Pdr12-GFP *snf1Δ*, Itr1-GFP and Itr1-GFP *snf1Δ* cells were grown in a glucose-containing medium and observed by fluorescence microscopy before and after 2DG treatment for 4h. Scale bar, 5 μm.
(TIF)

**S8 Fig. Snf1-GFP as a tool to follow Snf1 phosphorylation and function. A.** Serial dilutions of cultures of the WT, *reg1Δ*, *snf1Δ* and Snf1-GFP strains were spotted on synthetic complete medium containing either glucose or sucrose as a carbon source and grown for 3 days at 30°C. **B.** Total protein extracts of WT, Snf1-GFP, *reg1Δ*, *reg1Δ* Snf1-GFP and *snf1Δ* cells expressing

Mig1-Flag were prepared after overnight growth (exponential phase) in glucose medium (H: high glucose), after transfer to 0.05% glucose for 2h (L: low glucose) and after addition of glucose for 10 min (+Glc). Samples were immunoblotted using anti-Flag, anti-GFP and anti-pAMPK antibodies.MW: molecular weight marker.
(TIF)

**S9 Fig.** *HXK2* **deletion or** *DOG2* **overexpression prevent Hxt1 and Hxt3 endocytosis in response to 2DG. A.** Hxt1-GFP, Hxt1-GFP *hxk2Δ*, Hxt3-GFP and Hxt3-GFP *hxk2Δ*cells were grown in a glucose-containing medium and observed by fluorescence microscopy before and after 2DG treatment for 4h. Scale bar, 5 μm. **B.** Strains expressing Hxt1-GFP and Hxt3-GFP containing an empty plasmid (Ø) or plasmids allowing the overexpression of *DOG2* or its catalytic mutant *DOG2-DDAA* were grown in a glucose-containing medium and observed by fluorescence microscopy before and after 2DG treatment for 4h. Scale bar, 5 μm.
(TIF)

## Acknowledgments

We thank Agathe Verraes for technical assistance, Olivier Vincent (CSIC, Madrid, Spain) and Michel Becuwe (Harvard University, Cambridge, MA, USA) for sharing reagents and Alexandre Soulard (Université Claude Bernard Lyon 1, Lyon, France) for helpful discussions and critical reading of the manuscript. We also acknowledge the IJM ImagoSeine facility, member of IBiSA and the France-BioImaging infrastructure (https://anr.fr/ProjetIA-10-INBS-0004; ANR-10-INBS-04).

## Author Contributions

**Conceptualization:** Clotilde Laussel, Véronique Albanèse, Francisco Javier García-Rodríguez, Quentin Defenouillère, Sébastien Léon.

**Formal analysis:** Véronique Albanèse, Sébastien Léon.

**Funding acquisition:** Sébastien Léon.

**Investigation:** Clotilde Laussel, Véronique Albanèse, Francisco Javier García-Rodríguez, Alberto Ballin, Quentin Defenouillère.

**Methodology:** Clotilde Laussel, Véronique Albanèse, Francisco Javier García-Rodríguez, Alberto Ballin.

**Project administration:** Sébastien Léon.

**Resources:** Véronique Albanèse, Quentin Defenouillère.

**Supervision:** Sébastien Léon.

**Validation:** Clotilde Laussel, Véronique Albanèse, Francisco Javier García-Rodríguez.

**Visualization:** Clotilde Laussel, Véronique Albanèse, Sébastien Léon.

**Writing – original draft:** Sébastien Léon.

**Writing – review & editing:** Clotilde Laussel, Véronique Albanèse, Francisco Javier García-Rodríguez, Alberto Ballin, Sébastien Léon.

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
