## [Decision Letter · Decision Letter 0]

9 May 2022

Dear Dr LEON,

Thank you very much for submitting your Research Article entitled '2-deoxyglucose inhibits yeast AMPK signaling and triggers glucose transporter endocytosis, potentiating the drug toxicity' to PLOS Genetics.

The manuscript was fully evaluated at the editorial level and by independent peer reviewers. The reviewers appreciated the attention to an important topic but identified some concerns that we ask you address in a revised manuscript.

We therefore ask you to modify the manuscript according to the review recommendations. Your revisions should address the specific points made by each reviewer.

[LINK]

Yours sincerely,

Martin C. Schmidt

Guest Editor

PLOS Genetics

Gregory P. Copenhaver

Editor-in-Chief

PLOS Genetics

Reviewer's Responses to Questions

**Comments to the Authors:**

Reviewer #1: Clotilde Laussel and co-workers have used a combination of genetic, imaging and biochemical approaches to convincingly demonstrate that 2-deoxyglucose (2DG) inhibits AMPK signaling and thereby triggers a-arrestin (Rod1) mediated glucose transporter endocytosis in budding yeast, S.cerevisae. This process is detrimental for cells and potentiates the toxicity of 2DG.

On the mechanistic levels they show that the majority of 2DG enters the cells via the glucose transporters Hxt1 and Hxt3. The hexokinase Hxk2 rapidly converts 2DG into 2DG6P which then induces a transient activation of protein phosphatase 1 (PP1) and a concomitant inactivation of Snf1/AMPK. This results in the activation of the a-arrestin Rod1. Once activated, Rod1 enlists the HECT-type ubiquitin ligase Rsp5 to selectively downregulate a number of nutrient transporters, including the glucose transporters Hxt1 and Hxt3. The downregulation of Hxt1/3 exacerbates the metabolic blockade caused by 2DG (2DG6P) and ultimately leads to metabolic collapse and cell death. Interestingly loss of Rod1 causes 2DG tolerance. This might seem counterintuitive at first glance (more Hxt1/3 at the plasma membrane = more import of 2DG), however the authors have solved this conundrum.

They provide convincing evidence that in a rod1 mutant, Hxt1/3 are no longer downregulated in response to 2DG treatment. Surprisingly, this helps cells to detoxify 2DG, most likely by continuous import of glucose (resulting in metabolic competition) and by export of 2DG (or 2DG6P).

Overall, this is an interesting paper and should be published. The paper highlights the difference in AMPK signaling and 2DG action budding yeast and human cells, and it also offers a very interesting perspective on how a-arrestins dependent transporter regulation contributes to the mechanism of drug action.

Comments (most of which are really minor):

1. Fig. S4b – Please directly compare the 2DG resistance of rod1 mutants + Rod1-Flag and rod1 mutants + Rod1-HA on the same agar plates.

3. Please specify the lysine residues in Rod1 that are ubiquitinated by Rsp5 (not just Rod1KR).

4. Along the same lines, please specify the 12 serine residues in Rod1 that are phosphorylated by Snf1.

5. Fig. 2: Do mutations - Rod1KR, Rod1PYm and Rod1S12A also influence other Rod1 cargoes in addition to Ina1 (please also show for Hxt1/3).

5. Fig. 4: Do mutations – hxk2 mutants and overexpression of Dog2 (and the mutant version) also influence other Rod1 cargoes in addition to Ina1 (please also show for Hxt1/3).

6. Please check Snf1 phosphorylation at 4h 2DG treatment. Perhaps Snf1 only becomes hyperactive at later timepoints (4h)?

7. Fig. 5: I’m not sure that I fully can follow this argument: ‘Importantly, deletion of the gene encoding the amphiphysin orthologue RVS167, involved in endocytosis, did not result in 2DG resistance. Thus, it seems that only a partial endocytosis defect such as that observed in the rod1∆ (or rod1∆ rog3∆) mutant causes 2DG resistance.’ Could it be the rvs167 mutant is very sick anyways (in contrast to the rod1) and hence has trouble growing on 2DG? Also does loss of rvs167 really block Hxt1/3 endocytosis?

8. Figure 6B: The authors make a point that there is no difference in the intracellular 2DG levels between WT and rod1 mutants. I disagree – there is less 2DG in the rod1 mutant, but the difference is just not statistically significant. Also please do not use the * - *** system, just indicate the respective p-values (also for NS).

9. The authors argue that 2DG or 2DG6P is pumped out of the cells. Do they really provide evidence for that? Cloud they measure extracellular 2DG or 2DG6P measured (perhaps after chasing it?). If not please tone down the argument.

10. Fig. 6E: The FRET experiments indicate a slightly more glucose uptake in rod1 mutants after 4h of 2DG treatment and subsequent glucose addition. I suggest that the authors use the same approach to compare glucose influx in WT cells and rod1 mutants during the 4h 2DG treatment (provided of course the sensor is not activated by 2DG).

Reviewer #2: In the manuscript by Laussel et. al entitled, "2-deoxyglucose inhibits yeast AMPK signaling and triggers glucose transporter endocytosis, potentiating the drug toxicity", the authors investigated toxicity/resistance mechanisms of 2-deoxyglucose (2DG) in the model organism Saccharomyces cerevisiae. This is potentially an important piece of work that should be of interest to a broad audience as 2DG and its derivatives have anticancer potential and are used in cancer diagnostics.

Common resistance mechanism to 2DG involves constitutive activation of the 5’-AMP activated protein kinase (AMPK) Snf1 by mutating GLC7 and REG1 genes encoding subunits of the Protein Phosphatase 1 (PP1) complex, a negative regulator of Snf1. This leads to increased expression of DOG1 and DOG2 that encode phosphatases catalyzing dephosphorylation of 2DG-6-phosphate, a toxic metabolite of 2DG. It has been also shown that 2DG triggers alpha-arrestin (Rod1 and Rog3)-dependent endocytic degradation of Hxt1 and Hxt3 glucose transporters (O’Donnell et al., 2015, MCB). In addition, it was observed that 2DG treatment leads to increased phosphorylation and ubiquitination of Rod1 that requires Snf1 and PP1, respectively. This is consistent with the Snf1 role in preventing ubiquitination and activation of Rod1 when glucose is scarce. It was proposed that either 2DG increases activity of Snf1 leading to hyperphoshorylation and ubiquitination of Rod1 or Snf1-dependent phosphorylation of Rod1 is always inhibitory but 2DG simultaneously increases activity of PP1 towards Rod1 and/or Snf1. In sum, O’Donnell et al. proposed that 2DG toxicity largely results from glucose uptake inhibition caused by 2DG-induced internalization of glucose transporters.

In this work, the group of Sébastien Léon investigated the role of Snf1/PP1 pathway in Rod1-dependent internalization of glucose transporters Hxt1 and Hxt3 during 2DG treatment in more detail. The major novel findings are:

• 2DG addition triggers immediate but transitory dephosphorylation and inhibition of Snf1 that depends on PP1;

• 2DG treatment also induces rapid dephosphorylation of other Snf1 targets, such as Rod1 and Mig1, strongly suggesting upregulation of PP1 activity in the presence of 2DG;

• Phosphorylation of 2DG to 2DG-6-P by Hxk2 is required to promote PP1-independent dephosphorylation;

• 2DG toxicity largely results from Rod1-dependent internalization of glucose transporters and thus impairing both glucose uptake and detoxification of 2DG;

• 2DG addition triggers internalization of several plasma membrane proteins, also independently from Rod1.

Unfortunately, in my opinion, many conclusions are not convincingly supported by the presented data.

Major points:

1. The authors showed that 2DG treatment cause internalization of several plasma membrane proteins in both Rod1-dependent and independent way, including previously reported Hxt1 and Hxt3 (Figs 1, S2, S3, S5) . This is an interesting observations but this set of data is somehow detached from the rest of manuscript. It is not clear whether internalization of these plasma membrane proteins, except for Hxt1 and Hxt3, also contributes to 2DG toxicity. For Rod1-independent endocytosis, the authors could test involvement of other alpha-arrestins that might be also modified in response to 2DG by PP1-dependent dephosphorylation or other mechanisms. These data would be a good starting point for a separate story and should be deleted from the current manuscript.

2. One of major conclusions is that 2DG-6-P increases activity of PP1. However, this assumption is solely based on decrease in phosphorylation levels of PP1 targets such as Snf1 and Mig1. Such conclusion should be supported by direct measurements of PP1 enzymatic activity after addition of 2DG and/or testing increased physical interactions of PP1 with its substrates as an alternative hypothesis. In my opinion, presented results only support the notion that 2DG-6-P promotes/facililates PP1-dependent dephosphorylation of its targets by unknown mechanism.

3. Changes in phosphorylation status of Rod1 after 2DG treatment for 10 min are not convincing due to overlap of bands corresponding to phosphorylated and ubiquitinated forms of Rod1 (Fig. 2A-D). I do not notice any significant change in the intensity of upper band after CIP treatment in WT + 2DG, 10’ compared to –CIP sample (Fig. 2B). It is also difficult to judge changes in phosphorylation status of Rod1-KR devoid of ubiquitination site (Fig 2A).

4. It would be very informative to follow post-translational modifications of Rod1 and Mig1 as well as localization of Hxt1-GFP or Hxt3-GFP after 2DG addition at several time-points as it was done for Snf1 phosphorylation in Fig. 3C.

5. Based on data shown in Fig. 6A the authors concluded that deletion of ROD1 (and stabilization of hexose transporters) contributes to 2DG detoxification. However, it is not clear which substrate is detoxify, 2DG or 2DG-6-P? Similarly in the text, both forms are mentioned as subjected to detoxification (lines 305-306). Involvement of Hxt1 or Hxt3 in the detoxification process of 2DG is not demonstrated. In rod1Δ cells, efflux of 2DG or 2DG-6-P might be facilitated by other transporters stabilized in the plasma membrane. In general, more elaborated transport studies are needed to prove detoxification of 2DG/2DG-6-P and its mechanism.

Other comments:

1. Lines 141-142: “0.2% 2DG led to the degradation of PM-localized Tat1 and Hxt2 (Fig 1A), which was accompanied by their targeting to the vacuole (Fig 1B).”; it is better to say that “PM-localized Tat1 and Hxt2 were internalized and sorted to the vacuole for degradation. Panels Fig. 1A and 1B should be swapped.

2. Line 151: “2DG triggers a general endocytosis response”. This is unclear statement and not true as the complete plasma membrane proteome was not tested to conclude this.

3. Lines 158-159: “Altogether, these results indicate a prominent role for the ART protein Rod1 in 2DG-induced endocytosis.” This statement is too general as 2DG-induced internalization of many plasma membrane proteins are not dependent on Rod1. One could conclude that Rod1 is involved in 2DG-induced endocytosis of its targets.

4. Fig. 2A-B: please label bands of Rod1 as unmodified and Ub/P forms.

5. Fig. 2E-G and Fig. 4E-F: it is unclear to me why Ina1-GFP was used as a model cargo instead of Hxt1 or Hxt3, which are central to this paper.

6. Fig. 3A: Total proteins label is missing.

7. In Fig. 5D, the rod1Δ mutant with empty plasmid shows poor growth in the presence of 0.2% 2DG, whereas this strain shows a robust growth on 0.2% 2DG plates in other panels (Figs 2F, 5C-D).

8. Line 247: “ROD1 deletion leads to a partial 2DG resistance [14]”; How partial, compare to what? Please, specify.

9. Lines 257-258: ”Thus, it seems that only a partial endocytosis defect such as that observed in the rod1Δ (or rod1Δ rog3Δ) mutant causes 2DG resistance”. What is a partial endocytosis defect? This is rather a defect in endocytosis of certain proteins targeted by Rod1 or both Rog3 and Rod1.

10. Discussion lacks information about known mechanisms of PP1 regulation.

11. Page 17: The authors describe the method of Ina1-GFP purification for ubiquitylation analysis but there are no such data presented in this paper.

12. Figure 6 legend, line 956: 0,2% DG instead of 0.2% 2DG; line 967: 0,1% DG instead of 0.1% 2DG; why 0.1% 2DG was used instead of 0.2% in panel Fig. 6F?

13. Figure 5 and 6 legends: please clearly state for how many days the plates were incubated instead of saying 3-5 days.

Reviewer #3: The review is uploaded as an attachment.

**Have all data underlying the figures and results presented in the manuscript been provided?**

Reviewer #1: Yes

Reviewer #2: Yes

Reviewer #3: None

PLOS authors have the option to publish the peer review history of their article (what does this mean?). If published, this will include your full peer review and any attached files.

Reviewer #1: No

Reviewer #2: No

Reviewer #3: No

---

## [Decision Letter · Decision Letter 1]

20 Jul 2022

Dear Dr LEON,

We are pleased to inform you that your manuscript entitled "2-deoxyglucose transiently inhibits yeast AMPK signaling and triggers glucose transporter endocytosis, potentiating the drug toxicity" has been editorially accepted for publication in PLOS Genetics. Congratulations!

Yours sincerely,

Martin C. Schmidt

Guest Editor

PLOS Genetics

Gregory P. Copenhaver

Editor-in-Chief

PLOS Genetics

Comments from the reviewers (if applicable):

Reviewer's Responses to Questions

**Comments to the Authors:**

Reviewer #1: The authors have submitted a rigorously revised and improved version of their paper. They have addressed the most important questions, and clarified the major points. The new results further support the main conclusions put forward by the authors. This paper provides a new mechanism of how 2DG activates an a-arrestin ubiquitin ligase adaptor to drive glucose transporter endocytosis, which potentiates the drug toxicity in budding yeast. This is an important finding and should be published in plos genetics.

Reviewer #2: The authors have satisfactorily addressed my concerns and significantly improved the manuscript. Importantly, they toned down or deleted some conclusions. In general, very good revision.

**Have all data underlying the figures and results presented in the manuscript been provided?**

Reviewer #1: Yes

Reviewer #2: Yes

PLOS authors have the option to publish the peer review history of their article (what does this mean?). If published, this will include your full peer review and any attached files.

Reviewer #1: No

Reviewer #2: No

**Data Deposition**

http://datadryad.org/submit?journalID=pgenetics&manu=PGENETICS-D-22-00390R1

**Press Queries**

---

## [Editor Report · Acceptance letter]

8 Aug 2022

PGENETICS-D-22-00390R1 

2-deoxyglucose transiently inhibits yeast AMPK signaling and triggers glucose transporter endocytosis, potentiating the drug toxicity 

Dear Dr Léon, 

We are pleased to inform you that your manuscript entitled "2-deoxyglucose transiently inhibits yeast AMPK signaling and triggers glucose transporter endocytosis, potentiating the drug toxicity" has been formally accepted for publication in PLOS Genetics! Your manuscript is now with our production department and you will be notified of the publication date in due course.

With kind regards,

Olena Szabo

PLOS Genetics

On behalf of:
